# Accuracy of regional-to-global soil maps for on-farm decision making: Are soil maps "good enough"?

Jonathan J. Maynard[1], Edward Yeboah[2], Stephen Owusu[2], Michaela Buenemann[3], Jason C. Neff[1], and Jeffrey E. Herrick[4]

[1] Sustainability Innovation Lab, University of Colorado, Boulder, Colorado, United States of America
[2] CSIR-Soil Research Institute, Kwadaso, Kumasi, Ghana
[3] Department of Geography, New Mexico State University, Las Cruces, New Mexico, United States of America
[4] Jornada Experimental Range, Agricultural Research Service, United States Department of Agriculture, Las Cruces, New Mexico, United States of America

Correspondence to: Jonathan J. Maynard (jonathan.maynard@colorado.edu)

**Abstract.** A major obstacle to selecting the most appropriate crops and closing the yield gap in many areas of the world is a lack of site-specific soil information. Accurate information on soil properties is critical for identifying soil limitations and the management practices needed to improve crop yields. However, acquiring accurate soil information is often difficult due to the high spatial and temporal variability of soil properties at fine scales and the cost and inaccessibility of laboratory-based soil analyses. With recent advancements in predictive soil mapping, there is a growing expectation that soil map predictions can provide much of the information needed to inform soil management. Yet, it is unclear how accurate current soil map predictions are at scales relevant to management. The main objective of this study was to address this issue by evaluating the site-specific accuracy of regional-to-global soil maps, using Ghana as a test case. Four web-based soil maps of Ghana were evaluated using a dataset of 6,514 soil profile descriptions collected on smallholder farms using the LandPKS mobile application. Results from this study revealed that publicly available soil maps in Ghana lack the needed accuracy (i.e., correct identification of soil limitations) to reliably inform soil management decisions at the 1-2 ha scale common to smallholders. Standard measures of map accuracy for soil texture class and rock fragment class predictions showed that all soil maps had similar performance in estimating the correct property class. Overall soil texture class accuracies ranged from 8-14% but could be as high as 38-64% after accounting for uncertainty in the evaluation dataset. Soil rock fragment class accuracies ranged from 26-29%. However, despite these similar overall accuracies there were substantial differences in soil property predictions among the four maps, highlighting that soil map errors are not uniform between maps. To better understand the functional implications of these soil property differences, we used a modified version of the FAO Global Agro-Ecological Zone (GAEZ) soil suitability modelling framework to derive soil suitability ratings for each soil data source. Using a low-input, rain-fed, maize production scenario, we evaluated the functional accuracy of map-based soil property estimates. This analysis showed that soil map data significantly overestimated crop suitability for over 65% of study sites, potentially leading to ineffective agronomic investments by farmers, including cash-constrained smallholders.

## 1. Introduction

Site-specific soil information is urgently needed to address a variety of critical issues affecting agricultural systems, including soil fertility, erosion control, water management, and climate mitigation (Montanarella et al., 2015). Variability in both relatively static soil properties (such as clay content and depth) and current soil health (i.e., status of dynamic properties like fertility) is known to affect agricultural productivity. However, the lack of accurate information on soil physical and chemical properties has complicated or limited opportunities for smallholder farmers to improve soil health

through appropriate soil management practices (e.g., targeted fertilizer application). Smallholder farmers (i.e., farms <2 ha) cultivate 24% of agricultural land globally, yet generate 30-34% of the global food supply due to a higher percentage of agricultural production devoted to food crops (Ricciardi et al., 2018). With the development of improved crop varieties, smallholder farmers in many regions of the world have realized significant yield gains (e.g., Asia, Latin America) (Ritchie and Roser, 2013). While other areas, notably sub-Saharan Africa, have failed to realize appreciable yield increases due to

underlying biophysical constraints on crop production, principally soil infertility and the long-term depletion of soil nutrients (Sanchez, 2015).

   Smallholder farmers in Ghana are faced with a wide array of soil management challenges that affect the economic use of their soils. These challenges include low inherent fertility status, poor drainage, concretions and stoniness, shallow rooting depths, aluminium toxicity in acid soils, and susceptibility to both erosion and drought (Obeng, 1976; Obiri-nyarko, 2012). In

many of Ghana's major agricultural areas, increasing population pressure and inappropriate land use has contributed to extensive land degradation. Current agricultural yields in Ghana are far below their production potential. For example, average maize yield is around 1.7 tons per hectare, approximately one-quarter of the 6.0 tons/ha target set by Ghana's Ministry of Food and Agriculture (Chapoto and Tetteh, 2014). To overcome these challenges and increase crop yields, farmers must adopt improved production strategies, including the use of fertilizers, the planting of improved cultivars, and the adoption of good

agricultural practices (Fening, 2018). However, many soils in Ghana have severe constraints that limit the effectiveness of these production strategies, and without accurately identifying and addressing these soil limitations, smallholder farmers may fail to see a return on their investment. Accurate site-specific soil data could improve smallholder farmers' decisions and actions on sustainable agricultural practices and soil fertility management and thus lead to higher productivity potential.

   A major challenge to obtaining accurate site-specific soil data is the high spatial and temporal variability of soil properties

in many areas of the world. Soil variability results from differences in environmental factors (e.g., topography, geology, climate) that affect soil property formation over time (Bouma and Finke, 1993). The spatial scale and intensity at which these environmental factors vary determine the degree of soil variability within a landscape. Variation in certain soil properties can also result from the effects of management activities. For example, tillage and drainage of agricultural fields, crop rotation, application of fertilizers, and irrigation practices can all affect dynamic soil property values (e.g., organic matter, pH, plant

nutrient availability), particularly near the surface. Erosion and deposition can also affect what are typically considered static soil properties, such as texture and depth (Mulla and McBratney, 2001).

Ideally, smallholder farmers would be able to characterize the variability of their soils using laboratory-based physical and chemical analyses. In reality, high cost, limited access, and slow turnaround times have prevented most farmers from obtaining and using detailed soil laboratory information, while limited crop- and soil-specific knowledge have constrained

the use of this information. Soil maps have been widely viewed as at least a potential solution to this information gap, resulting in continued efforts to improve the spatial resolution and accuracy of soil map information (Brevik et al., 2015). While recent advancements in soil mapping allow for the prediction of soil information at management-relevant scales, the utility of those predictions depends on how accurately they portray fine-scale (i.e., 1-2 ha) soil variability. Failure to accurately characterize soil variability at the farm or field scale can severely limit the reliability of land suitability

assessments (i.e., fitness for a specific land utilization type, e.g., low-input, rain-fed wheat), and thus the ability to identify soil limitations and/or the conditions suitable for sustainable agricultural intensification.

Soil maps characterize spatial variability using either conventional or predictive soil mapping techniques. Conventional soil maps partition a landscape into finite circumscribed regions (i.e., soil map units), where boundaries are sharp lines delineating clear differences in soil types (Heuvelink and Webster, 2001). Conventional soil maps use empirical, expert-based

models to delineate the location and extent of soil types. These empirical models are often based on local geomorphology and vegetation patterns and validated by direct observation. The typical ranges of soil properties encountered for each soil type are established based on representative soil profiles and expert knowledge. In contrast, predictive soil maps characterize soil properties and classes (i.e., class probabilities) as continuous modelled values at a fixed grid cell resolution across a mapping area. Predictive soil maps are created from numerical or statistical models based on relationship among environmental

variables and soil properties or classes. These models often use legacy soil profile data and remotely sensed environmental covariates (e.g., slope, normalized difference vegetation index [NDVI]) that approximate soil forming factors (e.g., topography, climate, geology, vegetation). Predictive soil maps are driven by the modelling of spatial data and therefore limited by both the point data available for training/validation and the covariate data used for model development.

For many smallholder farmers, obtaining actionable soil information from soil maps is an attractive option. Multiple

sources of soil map information raise several important questions for end-users, including: How accurate are soil maps at my farm? Which soil map product is the most accurate? Can I use soil map information to inform my soil management decisions? Answers to these questions require an understanding of site-specific soil accuracy as it relates to both the relative accuracy of soil map information (i.e., compared to soil profile measurements) and the levels of soil map accuracy required for different land management applications (i.e., functional assessment).

This study evaluated the site-specific accuracy of four publicly available web-based soil maps of Ghana (Harmonized World Soil Database, World Inventory of Soil Property Estimates, SoilGrids250m, and iSDAsoil) using a dataset of 6,514 soil profile descriptions collected on smallholder farms using the LandPKS mobile application ('app') (Herrick et al., 2013). We evaluated three static soil properties (reflecting the long-term potential of the soil): soil texture class (USDA), rock fragment volume class, and soil depth (i.e., depth to bedrock). These properties directly affect agricultural production and can be used

to inform farmer decisions on a variety of management practices such as irrigation frequency and erosion control. They also

determine how susceptible soils are to declines in fertility and how responsive they are likely to be to different types and amounts of fertilizer and organic amendments such as compost and manure. We used standard measures of classification accuracy to assess the relative accuracy of each soil map. Furthermore, we conducted a global meta-analysis on the accuracy of field-based soil texture estimates so that we could derive estimation uncertainties for each USDA/FAO soil texture class, and thus account for potential uncertainty in our field-based soil texture class evaluation dataset. To help further contextualize these soil property differences, we used a modified version of the Global Agro-Ecological Zone (GAEZ) soil suitability modelling framework to derive soil suitability ratings for each soil data source (Fischer et al., 2008). Using a low-input, rain-fed, maize production scenario, we evaluated the functional accuracy of map-based soil property estimates relative to site-based measurements. The main objective of this study was to improve our understanding of differences in soil map products, the relation of these products to field observations, and the functional accuracy of soil map data for informing soil management recommendations.

## 2. Methods

### 2.1 Study Area

The study was conducted in Ghana, West Africa, within the Northern, Upper West, and Upper East regions in northern Ghana and the Western and Ashanti regions in southern Ghana (Fig. 1). The study area spans four agro-ecological zones, the Guinea Savannah and Sudan Savannah in the north and the Deciduous Forest and Wet Evergreen Rainforest in the south. The northern agro-ecological zones have a unimodal rainfall pattern with a mean annual rainfall of 1,100 mm, resulting in a single growing season from July to September. Agro-ecological zones in the south have a bimodal rainfall pattern and receive between 1500- and 2200-mm rainfall per annum, resulting in a major and minor cropping season. Soils in northern Ghana are predominantly Plinthisols and Planosols with smaller areas of Lixisols and Luvisols (Adjei-Gyapong and Asiamah, 2002; Awadzi and Asiamah, 2002). Soils in southern Ghana are predominantly Ferralsols and Acrisols, with smaller areas of Lixisols, Alisols, and Nitisols. Except for Luvisols in the north and Nitisols in the south, most soil types in Ghana have moderate-to-severe limitations for crop production, including low fertility (Acrisols, Alisols, Ferralsols, Lixisols), aluminum toxicity (Acrisols, Alisols, Planosols), shallow rooting depth (Plinthosols), high erosion risk (Alisols), and susceptibility to drought (Acrisols, Alisols, Ferrasols, Plinthosols).

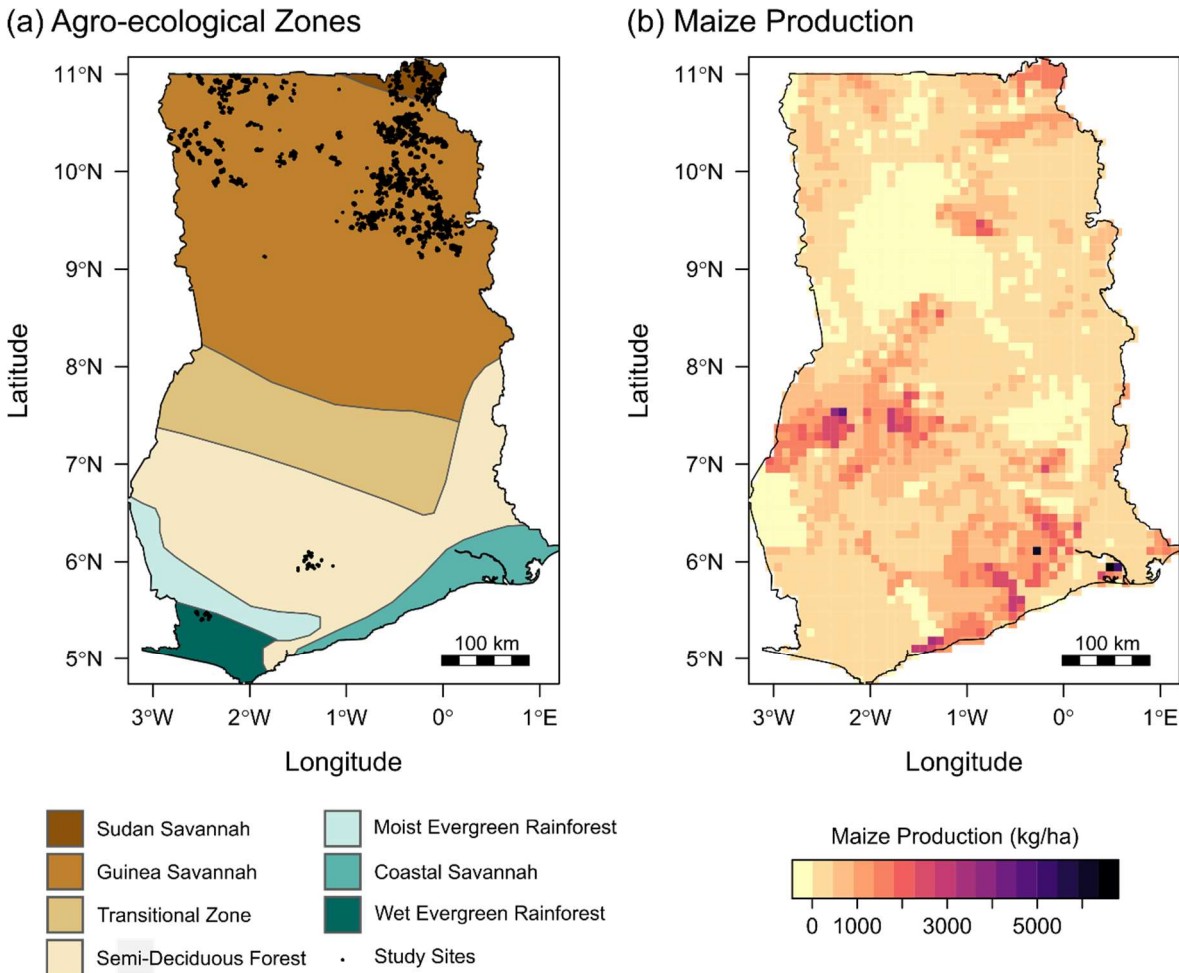

**Figure 1. Maps of Ghana showing (a) the location of LandPKS sampling sites underlain by Agro-ecological Zones and (b) the yield (kg./ha) of maize growing areas.**

## 2.2 Soil map acquisition and processing

Four soil mapping products were evaluated in this study: two conventional soil maps (i.e., Harmonized World Soil Database v1.21 [HWSD]; World Inventory of Soil Emission Potential 30 arc second map v1 [WISE]) and two predictive soil maps (i.e., SoilGrids250 v2 [SoilGrids]; iSDAsoil v1 [iSDA]) (Table 1). Conventional soil maps do not show the exact location of a soil type but instead display Soil Map Units (SMUs) representing distinct areas of a landscape composed of one or more soil types (i.e., soil map unit components). A common method for dealing with this spatial uncertainty is to assign any location within a SMU to its dominant soil component. In our comparisons of soil property values, we used the property values associated with

the dominant SMU component from the HWSD and WISE maps. In Ghana, HWSD is derived from the FAO-UNESCO Digital Soil Map of the World (DSMW) which has a map scale of 1:5,000,000 (translates to a spatial resolution of ~2.5 km). HWSD soil property data is derived using soil profile data from the WISE soil profile database and pedotransfer rules, producing two

aggregated soil depth intervals (0-30 and 30-100 cm) (Nachtergaele et al., 2009). The WISE soil map is a recent improvement upon HWSD, where an expanded WISE soil profile database and new pedotransfer rules were used to derive soil profile data at seven standardized depth intervals (0-20, 20-40, 40-60, 60-80, 80-100, 100-150, 150-200 cm). HWSD and WISE have identical spatial data (map scale: 1:5,000,000; spatial resolution: ~2.5 km) but differ in their soil property data (2 vs 7 depths for HWSD and WISE, respectively) (Batjes, 2016). Predictive soil mapping products (e.g., SoilGrids, iSDA) offer an

alternative to conventional soil maps by providing predictions of soil properties and classes at specific locations. SoilGrids is a global predictive soil map that predicts soil properties at a 250 m spatial resolution at six standard depths (0-5, 5-15, 15-30, 30-60, 60-100, and 100-200 cm) (de Sousa et al., 2020). iSDA is a predictive soil map of Africa that predicts soil properties at a 30 m spatial resolution at two standard depths (0-20, 20-50 cm) (Hengl et al., 2021). Soil map data was obtained from online repositories. Soil map predictions for sand, silt, and clay percentage, rock fragment volume, and depth to bedrock were

extracted from each map at all 6,514 sampling locations. For SoilGrids, depth to bedrock values were extracted from SoilGrids version 1.0 since no new map predictions were available for version 2.0. Among the soil data sources, the maximum prediction depth was shallowest for iSDA at 50 cm (Table 1).

 To facilitate comparison between the different soil data sources, we segmented each soil profile into 1 cm slices and then aggregated the slices (depth-weighted average) using a standard set of depth intervals (i.e., 0-10, 10-20, 20-50 cm). The

maximum soil depth for each data source was set at 50 cm to ensure all data sources had soil property values at each depth interval in our comparison (Fig. 2). The segmenting algorithm was implemented using the 'aqp' package for R (Beaudette et al., 2013). For each reaggregated soil depth interval, we calculated soil texture class based on USDA texture classes and rock fragment volume class based on the LandPKS rock fragment class intervals (i.e., 0-1, 1-15, 15-35, 35-60, and >60%).

**Table 1. Soil data sources in Ghana**

| Soil Data | Version | Spatial Extent | Scale / Resolution† | Map-unit | Spatial Support | Depth support |
|---|---|---|---|---|---|---|
| HWSD | 1.21 | Global | 1:5,000,000 | Polygon | Area | 2 layers: 0-30, 30-100 cm |
| WISE | 1.0 | Global | 1:5,000,000 | Polygon | Area | 7 layers: 0-20, 20-40, 40-60, 60-80, 80-100, 100-150, 150-200 cm |
| SoilGrids | 1.0, 2.0 | Global | 250 m | Raster | Point | 6 layers: 0-5, 5-15, 15-30, 30-60, 60-100, 100-200 cm |
| iSDA | 1.0 | Africa | 30 m | Raster | Point | 2 layers: 0-20, 20-50 cm |
| LandPKS | 2.1.0 | Field | <1 m | Point | Point | 5 layers: 0-1, 1-10, 10-20, 20-50, 50-70 cm |

†Map scale of 1:5,000,000 translates to a spatial resolution of approximately 2.5 km.

**2.3 Field data collection**

Soil profiles were sampled as part of two different monitoring and evaluation (M&E) surveys of smallholder farmers in Ghana: USAID's Feed the Future (FTF) project (Northern Ghana) and a World Bank funded research project, Map to the Future (M2F) (Southern Ghana). The FTF project used a cross-sectional multi-stage cluster sampling design, using probability proportional to size sampling to select smallholder farms (USAID, 2013). At each selected farm a single representative site (i.e, visually assessed to represent the average biophysical condition) was selected at each farm for soil sampling (farms/soil profiles=6,289).

The M2F project used a conditioned Latin hypercube sampling design (cLHS) to select a subset of smallholder farmers participating in an agricultural advisory pilot project (FarmGrow: Daniel et al., 2020). Baseline agronomic information (e.g., agricultural practices, soil condition, annual yield) was used to stratify the cLHS subsampling. At each selected farm in the M2F project, three soil profiles were sampled from each farm field, with sampling locations chosen by the farmer to reflect within-field soil variability (farms=75, soil profiles=225).

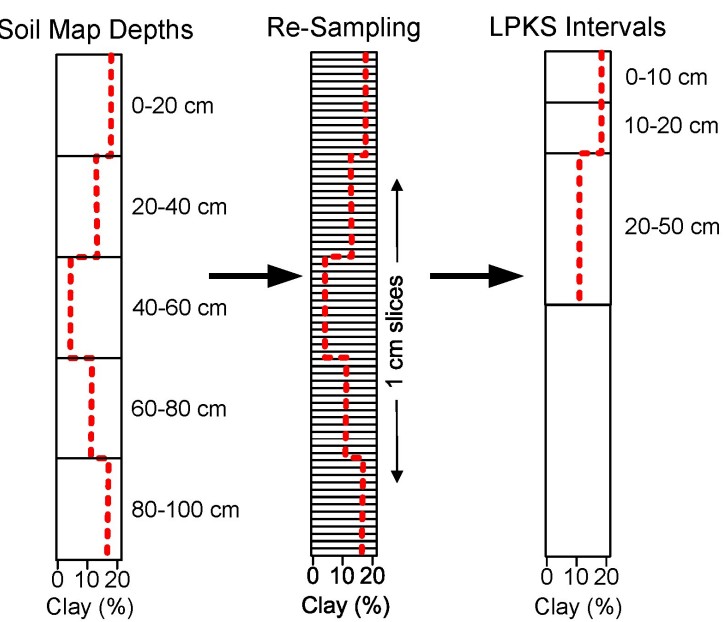


**Figure 2. Soil profile slicing and aggregation method for converting contrasting soil sampling depths to the standard LandPKS sampling depths for all properties, showing an example of clay percentage.**

Data collection was performed using the LandPKS mobile app by field crews following standard sampling protocols (Zalisk et al., 2018; https://landpotential.org/knowledge-hub/). This involved sampling of soils by either hand auger (northern Ghana) or from soil pits (southern Ghana) at 5 standard depth intervals (i.e., 0-1, 1-10, 10-20, 20-50, 50-70 cm). Soil samples were passed through a 2 mm sieve and analysed for soil texture (USDA/FAO textural classification) using the hand texturing method (Schoeneberger et al., 2012) and rock fragment volume (i.e., volume percent of rock fragments >2mm) class (i.e., 0-1%, 1-15%, 15-35%, 35-60%, >60%) using visual estimates (USDA-NRCS, 2020). Depth to bedrock was also recorded if encountered within the 70 cm sampling depth.

While most soil map predictions are derived from laboratory-based property measurements (e.g., HWSD, WISE, SoilGrids), several recent studies have shown field-estimated soil property values can produce relatively accurate estimates when compared to laboratory measurements (Salley et al., 2018; Vos et al., 2016a). For example, Salley et al. (2018) reported that professional soil scientists and field technicians correctly estimated laboratory-determined texture classes for 66% and 41% of samples, respectively. And when a 'correct' prediction also included adjacent textural classes, accuracies increased to 91% and 78% for professionals and field technicians, respectively (Salley et al., 2018). The compatibility of these different measurement methodologies was recently demonstrated with the iSDA soil maps which used both laboratory and field-based measurements to predict soil texture and rock fragment volume (Hengl et al., 2020).

All data recorded in the LandPKS app were synchronized to a cloud-based data storage system. Soil profile data were downloaded from the LandPKS data portal (https://landpotential.org/data-portal/, accessed Nov 6, 2020). Quality control filtering was performed on LandPKS data to remove incomplete sites. This included removing sites with missing soil property data and sites that were not sampled at all 5 depth intervals.

**2.4 Soil evaluation datasets**

In developing our soil map evaluation procedure, we identified three potential issues with our evaluation datasets that needed to be addressed: (1) independence from the soil maps being evaluated, (2) spatial support of the evaluation data, and (3) accuracy and precision of the evaluation measurements. The evaluation data from the FTF project (6,289 sites) in Northern Ghana was used as part of the iSDA model calibration/training dataset, and therefore could not be used for independent evaluation of the iSDA map predictions (Hengl et al., 2020). Additionally, only one location was sampled within each smallholder farm for the FTF project, thus requiring all evaluation to be done at the point support (i.e., individual site value vs. predicted map value). Although both predictive soil maps used a point prediction support (i.e, each soil measurement represents a single point on the ground), previous studies have shown that validating at a point-support can underrepresent a maps prediction quality, and it is therefore preferable to validate at larger spatial supports (e.g., block support) (Bishop et al., 2015; Piikki et al., 2021). The M2F study sites, although a considerably smaller dataset (n=225) and concentrated in southern Ghana, were not used in the iSDA model and each farm was sampled at three locations allowing for both an independent accuracy assessment of iSDA predictions and for accuracy assessments at both point and field support. To address the issues of evaluation independence and evaluation support, we evaluated three datasets: (1) the entire dataset at point-support [FTF-

M2F-PS: 6,514 study sites], (2) the M2F dataset a point-support [M23F-PS: 225 study sites], (3) the M2F dataset at field-support [M2F-FS: 75 farms]. The final issue relates to accuracy and precision of the evaluation measurements. The FTF and M2F datasets contain field estimates of soil texture class and rock fragment volume. Rock fragment volume measurements are commonly estimated in the field using visual or ocular assessment techniques and all of the soil maps evaluated in this study incorporate field-based soil rock fragment data in their map predictions (Ribeiro et al., 2020). Additionally, we used broad rock fragment volume classes which minimized any potential methodological differences between our evaluation dataset and the data used to generate soil map prediction. Soil map predictions of texture, however, are typically made using laboratory measurements, and thus our evaluation of soil map predictions using field texture measurements may bias our accuracy assessment. Therefore, to evaluate the compatibility of field and laboratory texture data we conducted a global meta-analysis on the accuracy of field-based soil texture estimates.

## 2.5 Global meta-analysis of field-based soil texture estimation uncertainty

To account for potential uncertainty in our soil texture evaluation dataset, we conducted a global meta-analysis on the accuracy of field-based soil texture estimation. Data was compiled from 10 studies that were reported in 7 peer-reviewed publications. Eight of the 10 studies used the USDA/FAO texture classification (Foss et al., 1975; Levine et al., 1989; Post et al., 1986; Rawls and Pachepsky, 2002; Salley et al., 2018), while one used the Australian (Northcote) classification systems (Minasny et al., 2007) and the other used the French (AISNE) classification systems (Richer-de-Forges et al., 2022). Data from the Australian and French studies were transformed to the USDA/FAO system using the 'texture' R package (Moeys, 2018). Additional details on the soil texture transformation are found in the Supplemental Materials. The final compiled dataset contained 269,181 hand-texture and corresponding lab texture measurements. The majority of these texture measurements were from the US (USDA-NRCS: 228,715) (Salley et al., 2018), followed by Australia (ASRIS/Queensland Government: 17,979) (Minasny et al., 2007), France (17,388) (Richer-de-Forges et al., 2022), and 5,099 samples from the remaining 7 studies, with sample sizes ranging from 154 to 1,724 (Table 2). Two of the studies only reported texture class accuracies, another two reported texture class accuracies along with additional information on misclassified classes, and the remaining 6 studies provided soil texture class error matrices. For the 8 studies with misclassification information (error matrices or reported misclassifications), we calculated texture class accuracies that accounted for class adjacency ($PA_{adj}$). Weighted mean field texture class accuracies (PA and $PA_{adj}$) were calculated for each texture class, weighted by each studies contribution to the texture class sample size. To estimate the potential error associated with the use of field estimated texture classes relative to laboratory measurements, we estimated an overall accuracy for each evaluation dataset by calculating a weighted average (mean ± standard deviation) of the individual texture class accuracy estimates, weighted by the texture class distributions for each evaluation dataset.

**2.6 Soil map accuracy assessment at field-support**

To calculate accuracy measures at field-support we need to compare the average of site values within a field to the average of all predicted map values within a field. For the study sites from the M2F research project in southern Ghana, the exact boundaries of each field were not available. Consequently, we approximated the area of each field by creating a convex hull around each set of sampling points (n=3) within a field and then applied a 10 m buffer around the perimeter of each delineated area. Using the approach described by Bishop et al. (2015), each buffered area was then discretised into a 10 m grid, with the center of each grid converted to a point and used for extracting soil map predictions. A 10 m grid was chosen to ensure representative sampling of the soil maps across all grid resolutions. The values extracted at each point where then averaged, giving an approximate area weighted average for each sub-field delineation. The average measured field values were obtained by averaging values from the three soil profiles within each field.

**2.7 Soil map evaluation methods**

We evaluated the relative and functional accuracy of the soil maps using two different methods, (1) matching of soil property classes (relative accuracy), and (2) matching of crop-specific GAEZ soil suitability ratings (functional accuracy).

**2.7.1 Soil property class match**

The soil property class match approach applies an exact matching criterion where the measured soil property class at each site and soil depth is compared to the predicted property class in each soil map. Because this approach requires an exact match it can result in a high rate of misclassification among similar soils and therefore provides a conservative measure of map accuracy. We addressed this with a second measure that also considers all adjacent property classes to be correct. For this method, we evaluated map accuracy for soil texture and rock fragment volume classes.

Map performance was evaluated using overall map accuracy, adjacent-overall accuracy, producer's accuracy, user's accuracy, and balanced error rate. Overall map accuracy (OA) is the proportion of all observation points at which the map predicts the correct soil property class (i.e., soil texture class, rock fragment volume class). Adjacent-overall accuracy ($OA_{adj}$) includes all property classes adjacent to the correct class as a 'correct' prediction. Producer's accuracy (PA) and user's accuracy (UA) are calculated separately for each class. PA is the probability that a ground reference test sample is classified correctly in the map (e.g., what proportion of clay loam reference samples were correctly classified on the map [*True positive/(True positive + False Negative)*]). UA is the probability that a sample from a map actually represents that category on the ground (e.g., what proportion of reference samples mapped as clay loam were truly clay loam [*True positive/(True positive + False Positive)*]. The balanced error rate (BER) is the average of the errors in each property class. This includes both errors associated with a failure to predict the correct class (false negative or Type II error), as well as the error associated with allocating a sample or site to the wrong class (false positive or Type I error). BER is calculated by taking the average of the false negative

rate (FNR) (i.e., [*False negative /(False negative + True positive)*]) and the false positive rate (FPR) (i.e., [*False positive/(False positive + True negative)*]):

$$BER \ = \ (FPR \ + \ FNR)/2, \tag{1}$$

Using the average of the FNR and FPR, BER is sensitive to problems of class imbalance, where models that overpredict the dominant class will receive a higher value (e.g., close to 1.0).

### 2.7.2 Global Agro-Ecological Zone soil suitability

Assessing map accuracy based on the soil property class match rate fails to account for when the predicted class is functionally similar to the measured class. In other words, sometimes misclassification of a soil property simply does not matter much for

management. For example, a sand texture misclassified as a loamy sand would be functionally more similar than a sand texture misclassified as a sandy clay. To account for these relative differences, we evaluated the functional similarity between data sources using a simplified version of the GAEZ soil suitability modelling framework. The GAEZ framework, developed by the Food and Agriculture Organization of the United Nations (FAO) and the International Institute for Applied Systems Analysis (IIASA), uses soil data and detailed agronomic knowledge to quantify land productivity and crop-specific agronomic

potential (Geze Toth, Bartosz Kozlowski, Sylvia Prieler, 2012). GAEZ soil suitability calculations follow a two-step approach, where (1) cropping system-specific responses (i.e., unique combination of crop type, management level, and water supply) to individual soil properties are combined into soil quality ratings, and (2) individual soil quality ratings are combined to calculate management-specific soil suitability ratings. The soil suitability ratings serve as a functional metric for comparing differences in the soil property predictions between the different soil maps. The GAEZ soil quality framework uses multiple soil properties

to calculate each of the soil quality indices, including: soil nutrient availability ($SQ_N$) = $f$(soil texture, organic carbon, pH, and total exchangeable bases); soil rooting conditions ($SQ_R$) = $f$(soil depth, soil phases); and soil workability ($SQ_W$) = $f$(soil depth, texture, rock fragments, soil phases, vertic soil properties).

Our modified GAEZ framework used a low-input, rain-fed, maize production scenario to translate soil property information at each site into crop-specific soil suitability ratings for each soil data source. We calculated simplified soil quality

indices using soil texture class, rock fragment class, and soil depth as input properties. We used maize as our modelled crop due to its widespread cultivation throughout Ghana (Fig. 1) and our selection of input soil properties was limited by those properties common to all data sources. Soil property values at each site were used to calculate three different soil quality indices (SQs): soil nutrient availability ($SQ_N$), soil rooting conditions ($SQ_R$), and soil workability ($SQ_W$). Each soil quality index has its own unique set of soil properties ratings based on their relative influence. SQs: $SQ_N$ = soil texture, organic carbon,

pH, and total exchangeable bases; $SQ_R$ = soil depth, soil phases; $SQ_W$ = soil depth, texture, rock fragments, soil phases, vertic soil properties. We limited the calculation of SQs to the soil properties common among all five soil sources, which were soil texture, rock fragments, and soil depth.

The natural availability of soil nutrients is critical for crop productivity in low-input farming systems. Soil texture class was used as an indicator of rock-derived nutrient availability, with finer textured soils (e.g., clay) typically having higher

nutrient availability than coarse textured soils (e.g., sand). The rock-derived nutrients include phosphorus, micro-nutrients, and base cations, and many of these nutrients are associated with specific mineralogy and tend to be less concentrated in sandy soils (Sollins et al., 1988). In contrast, nitrogen is substantially influenced by nitrogen fixation and soil organic matter content. Soil nutrient availability was calculated as:

$$SQ_N = STR, \tag{2}$$

where $STR$ is the soil texture class rating.

Soil rooting condition assesses the effective soil depth and volume for crop roots by accounting for the effects of soil depth, soil texture, and rock fragments volume. The soil rooting condition index was calculated as:

$$SQ_R = SDR * min\,(STR, RFR)\,, \tag{3}$$

where $SDR$ is the soil depth rating, $STR$ is the soil texture rating, and $RFR$ is the rock fragment rating.

Soil workability or ease of tillage is affected by both physical hindrances to cultivation (e.g., bedrock, rock fragments) and limitations imposed by soil texture. Soil workability was calculated as:

$$SQ_W = \frac{X_{jo} + 0.5 \sum_{j \neq jo} X_j}{2}, \tag{4}$$

where $X$ is the soil property rating (i.e., SDR, STR, RFR), $jo$ denotes the soil property with the lowest rating such that: $SR_{jo} \leq SR_j, j=1{:}3$.

The three soil quality indices were combined to calculate the soil suitability rating (SR):

$$SR = SQ_N * SQ_R * SQ_W, \tag{5}$$

## 3. Results

### 3.1 Soil Property Distributions

While the spatial distribution of surface soil texture classes differed among the four soil maps, they all displayed a general trend of coarser soil textures in the north and finer soil texture in the south (Fig. 3). Furthermore, all four maps showed similarities in the relative distribution of certain soil texture classes, with sandy loam, loam, sandy clay loam, and clay loam dominant within most maps (Fig. 4a-d). The WISE soil map predicted the highest diversity of soil texture classes (i.e., classes ≥1% map area) with six texture classes, followed by HWSD and SoilGrids with five texture classes, and iSDA with four texture classes (Fig. 4). LandPKS field-based measurements spanned the widest range of textures, with a total of 11 classes (Fig. 4e). This is not surprising given the natural variability of soil texture at fine spatial scales. For WISE, HWSD, and SoilGrids, both the diversity of soil property classes and their relative distributions across the 6,514 study sites (Fig.4f-h), which comprised a total of 19,542 soil layers, were similar to the soil property maps (Fig. 4a-c). In contrast, ISDA soil property distributions were markedly different at the study sites (Fig. 4i), exhibiting a higher diversity of property classes relative to their depth-wise areal distributions across Ghana (Fig. 4d), likely a result of the influence of the FTF sites on iSDA model predictions. This contrasted

with the M2F dataset (independent of iSDA model predictions) where the distribution of iSDA texture predictions was more closely aligned with the depth-wise areal distributions across Ghana.

LandPKS sites are predominantly coarse textured soils with 71% of soil layers classified as sandy loam or coarser. In contrast, SoilGrids predicted only 8% and HWSD 21% of soil layers as sandy loam or coarser. WISE was more similar with 51% of soil layers, and the predictions from iSDA the most similar with 77% of soil layers classified as sandy loam or coarser. Figure 5 shows the distribution of soil rock fragment classes for the LandPKS sites and corresponding soil map values. LandPKS sites showed a range of soil rock fragment classes in the FTF-M2F dataset, with an almost equal distribution among the first four classes. WISE and SoilGrids had high percentages in several of the higher rock fragment classes, which more closely aligned with LandPKS values, while HWSD predicted low rock fragments across the majority of sites (97% in the 1-15% class). For the M2F sites, LandPKS values were spread across multiple rock fragment classes (Fig. 5j), in contrast to HWSD, WISE and ISDA which predicted almost all sites in the 1-15% class (Fig. 5).

## 3.2 Uncertainty of field-based soil texture estimates

Results from the meta-analysis of field-based soil texture uncertainty are shown in Table 2. In general, results from these studies showed that very coarse (e.g., sand) and very fine (e.g., clay) soil texture classes were estimated with higher accuracies and lower variability relative to medium texture classes (e.g., loam). For example, average accuracies were 73±7% for sand and 74±15% for clay, compared to 41 ±19% for sand clay loam among 10 different studies (Foss et al., 1975; Levine et al., 1989; Minasny et al., 2007; Post et al., 1986; Rawls and Pachepsky, 2002; Richer-de-Forges et al., 2022; Salley et al., 2018). Additionally, texture classes with the lowest sample sizes generally had lower estimation accuracies and higher variability, for example the sandy clay (28 ±22%) and silt (19 ±16%) classes which each represent ~1% of compiled dataset.

**Table 2 Accuracy of field estimated USDA/FAO soil texture classes.**

| Research Study | Ns† | S | LS | SL | SC | SCL | L | SiL | CL | SiCL | SiC | C | Si |
|---|---|---|---|---|---|---|---|---|---|---|---|---|---|
| | | **Soil Texture Class Accuracy (%)** | | | | | | | | | | | |
| Foss et al., 1975 | 598 | 74 | 35 | 68 | – | 44 | 36 | 63 | 18 | 19 | – | 73 | – |
| | | [89] | [73] | [83] | – | [44] | [58] | [73] | [46] | [44] | – | [80] | – |
| Post et al., 1986 | 900 | 86 | 46 | 69 | – | 30 | 26 | 59 | – | 47 | 36 | – | – |
| | | [97] | [85] | [98] | – | [87] | [81] | [92] | – | [95] | [84] | – | – |
| Levine et al., 1989 (WK3) | 1,725 | 81 | 46 | 47 | – | 19 | 21 | 19 | 50 | 41 | 47 | 48 | – |
| | | – | – | – | – | – | – | – | – | – | – | – | – |
| Levine et al., 1989 (EOS) | 810 | 96 | 55 | 57 | – | 35 | 27 | 17 | – | 33 | 31 | 47 | – |
| | | – | – | – | – | – | – | – | – | – | – | – | – |
| Rawls and Pachepsky, 2002 | 576 | 24 | 24 | 73 | – | 15 | 46 | 64 | 16 | 44 | 32 | 58 | – |
| | | [60] | [100] | [99] | – | [93] | [97] | [99] | [100] | [84] | [87] | [82] | – |
| Minasny et al., 2007 | 17,979 | 62 | 33 | 24 | 13 | 14 | 29 | 10 | 6 | 73 | 6 | 64 | 14 |
| | | [94] | [91] | [67] | [51] | [53] | [70] | [30] | [24] | [93] | [84] | [93] | [55] |
| Salley et al., 2018 (USDA) | 228,715 | 73 | 42 | 61 | 28 | 49 | 58 | 81 | 57 | 70 | 59 | 74 | 16 |
| | | [88] | [94] | [93] | [90] | [91] | [94] | [99] | [86] | [95] | [93] | [92] | [94] |
| Salley et al., 2018 (BLM) | 154 | 43 | 44 | 55 | – | 25 | 5 | 26 | 33 | 44 | – | 40 | – |
| | | [71] | [100] | [89] | – | [100] | [65] | [77] | [83] | [78] | – | [40] | – |
| Salley et al., 2018 (Namibia) | 336 | 45 | 32 | 24 | – | 0 | 0 | 10 | – | – | – | – | – |
| | | [80] | [59] | [47] | – | [100] | [0] | [90] | – | – | – | – | – |
| Richer-de-Forges et al., 2022 | 17,388 | 95 | 82 | 66 | 76 | 73 | 61 | 61 | 68 | 66 | 75 | 82 | 92 |
| | | [100] | [99] | [98] | [98] | [97] | [92] | [96] | [95] | [96] | [98] | [99] | [99] |
| **PA** | | 73 ±7 | 45 ±12 | 57 ±11 | 28 ±22 | 41 ±19 | 54 ±10 | 79 ±8 | 57 ±7 | 69 ±4 | 59 ±7 | 74 ±4 | 19 ±16 |
| N$_{TC}$† | 269,181 | 15,116 | 9,805 | 36,425 | 3,033 | 13,385 | 36,470 | 55,292 | 21,058 | 33,004 | 16,094 | 27,184 | 2,314 |
| **PA$_{adj}$** | | 89 ±3 | 94 ±3 | 91 ±8 | 71 ±21 | 81 ±18 | 91 ±8 | 98 ±6 | 86 ±8 | 95 ±3 | 93 ±2 | 93 ±2 | 80 ±19 |
| N$_{TC-adj}$† | 266,646 | 15,060 | 9,469 | 35,702 | 3,033 | 13,118 | 35,895 | 55,213 | 21,012 | 32,833 | 15,992 | 27,004 | 2,314 |

† N$_S$ , sample size of individual studies; N$_{TC}$, sample size of individual soil texture classes used to calculate PA, N$_{TC-adj}$, sample size of individual soil texture classes used to calculate PA$_{adj}$

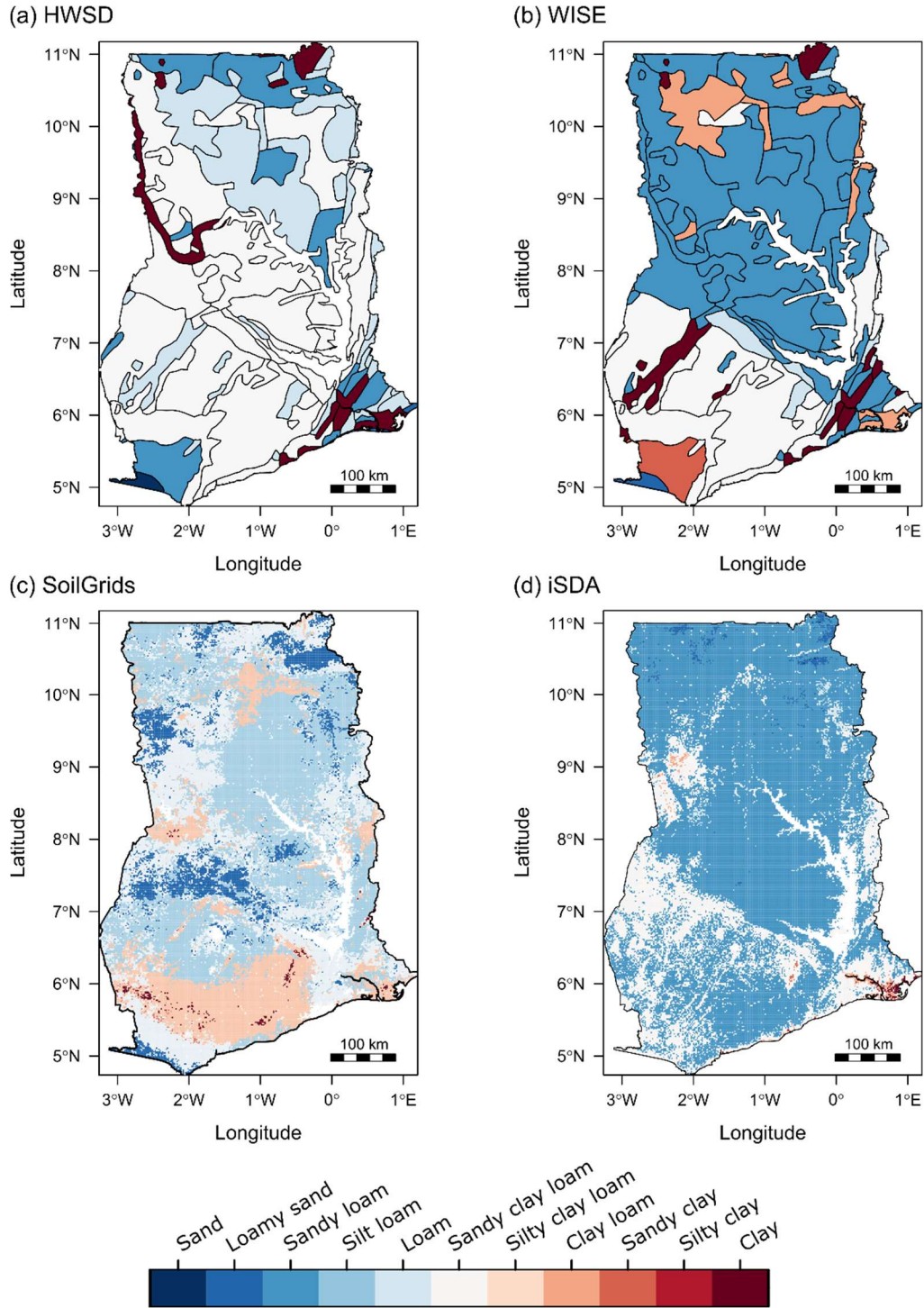

**Figure 3. Soil map comparison of surface soil (0-10 cm) texture classes. Texture classes are ordered by mean particle size diameter.**

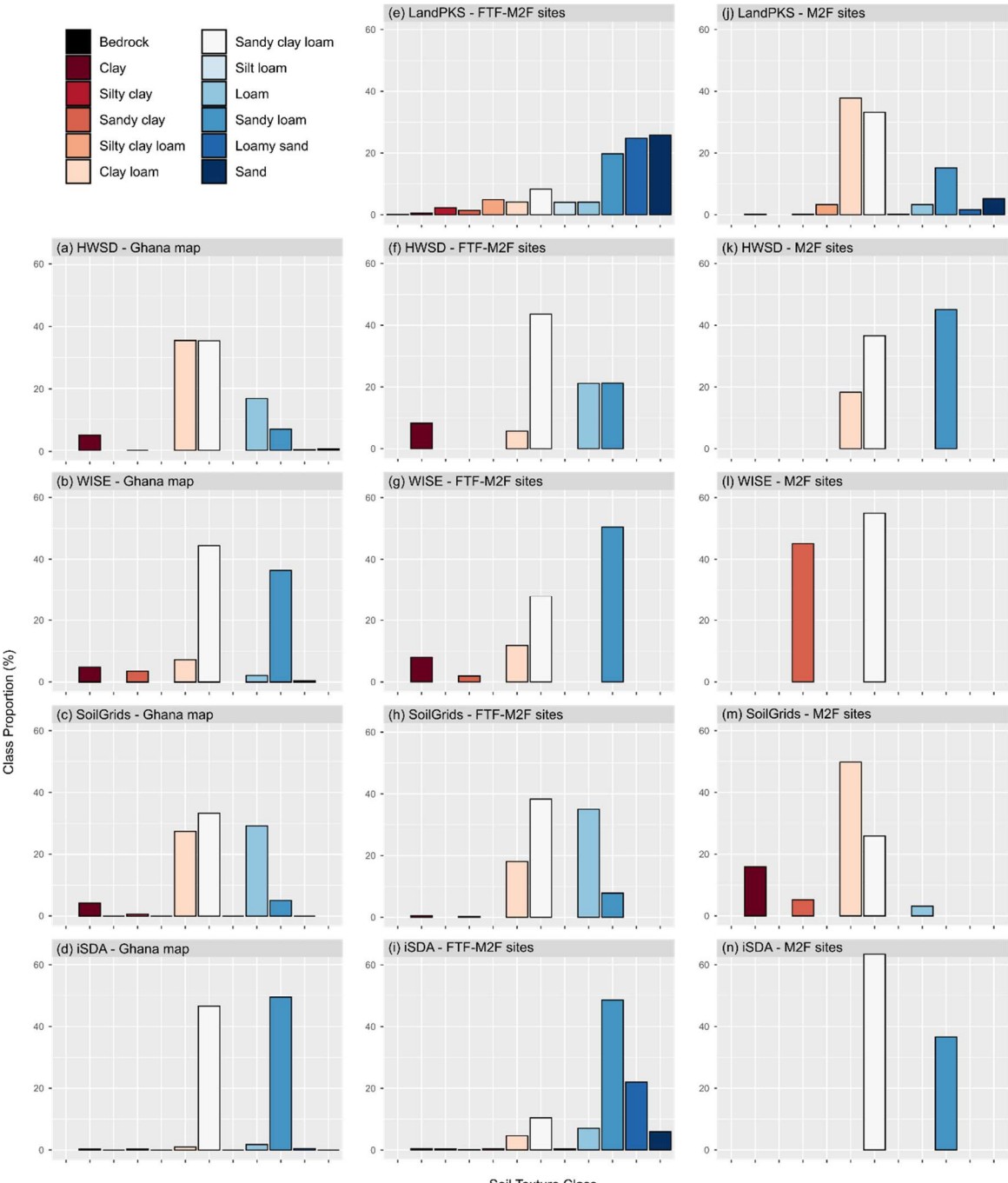

**Figure 4. Distribution of soil texture classes based on (a-d) areal map coverage across Ghana , (e-i) distribution across the FTF-M2F dataset, and (j-n) distribution across the M2F dataset (point-support) at LandPKS depths. Class proportions account for all LandPKS soil depths ≥ 50 cm (i.e., 0-10, 10-20, 20-50 cm), with equal weight assigned to each depth interval regardless of its total depth.**

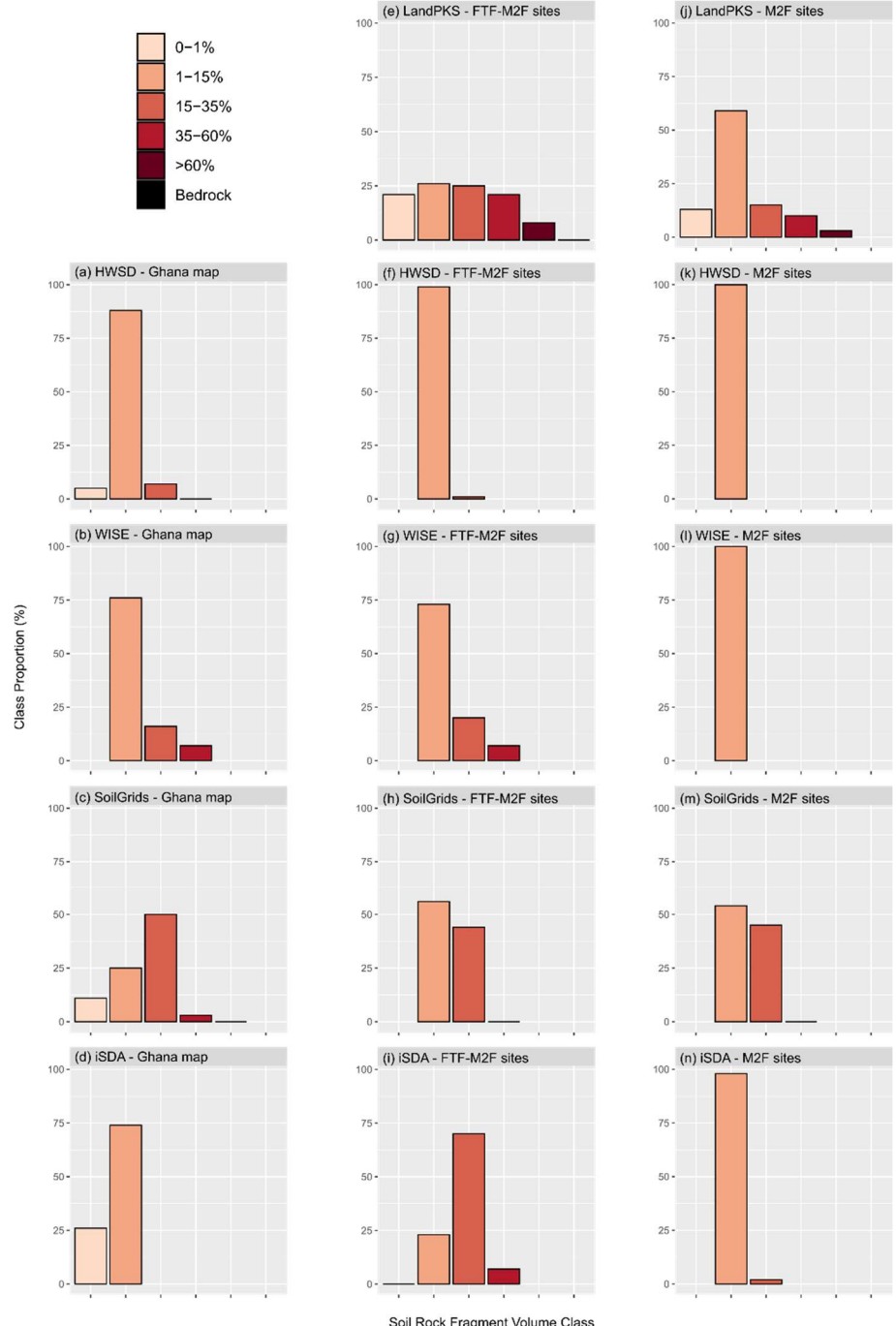

**Figure 5. Distribution of soil rock fragment volume classes based on (a-d) areal map coverage across Ghana, (e-i) distribution across the FTF-M2F dataset, and (j-n) distribution across the M2F dataset (point-support) at LandPKS depths. Class proportions account for all LandPKS soil depths ≥ 50 cm (i.e., 0-10, 10-20, 20-50 cm), with equal weight assigned to each depth interval regardless of its total depth.**

### 3.2 Evaluation of Soil Map Accuracy

### 3.2.1 Soil Property Class Match

Overall accuracy of soil property maps was low, with soil texture classes ranging from 9-14% for FTF-M2F and 15-35% for M2F and soil rock fragment classes ranging from 26% to 29% for FTF-M2F and 33-59% for M2F (Table 3). While overall accuracies were slightly higher in southern Ghana (M2F dataset) for both texture and rock fragments, these higher accuracies were likely due to the overprediction of the dominant classes. For example, in the M2F dataset, WISE, SoilGrids and iSDA soil maps predicted 99-100% of the sites in the 1-15% rock fragment class which was the most dominant measured class (i.e, 59% of sites; Fig. 5j). This resulted in higher model accuracy but low sensitivity for all other classes (Figs. 6,7). The balanced error rate was high across all maps, ranging from 75% to 95% for soil texture class and 79% to 83% for soil rock fragment class.

Due to potential inaccuracies in field estimated soil texture classes, we estimated the agreement between field and laboratory measured texture classes using a global meta-analysis of soil field texture measurement accuracy. Estimated agreement (± standard deviation) between field and laboratory values was 58 ±10% and 53 ±11% for the FTF-M2F and M2F evaluation datasets, respectively. Estimated agreement increased to 90 ±5% for FTF-M2F and 86 ±11% for M2F when allowing for class adjacency ($OA_{adj}$). Thus, the potential error rate of our soil texture class evaluation datasets ranged from 32-52% for FTF-M2F and 36-58% for M2F for an exact match, which decreases to 5-15% for FTF-M2F and 3-25% for M2F for adjacent class matches. Using these potential error rates, we adjusted the soil texture map accuracies to reflect potential accuracy if all error in the evaluation dataset corresponded to the misclassifications in the map dataset. Moderate-to-high uncertainty in both our evaluation datasets and the soil map predictions makes it difficult to assign a realistic upper bound for the potential accuracies of soil map predictions (i.e., map accuracies adjusted for evaluation data uncertainty: OA-P). However, these upper bounds should not exceed the potential adjacent texture class accuracies ($OA_{adj}$-P), which have more constrained uncertainty estimates and thus more realistic estimates of potential map accuracy. Thus, whenever the estimated upper bound of OA-P exceeded the upper bound of $OA_{adj}$-P it was truncated to match $OA_{adj}$-P (Table 3). These adjustments resulted in potential soil map texture class accuracies ranging from 41-64% for an exact match and 44-64% for an adjacent match for the FTF-M2F dataset, and 52-93% for an exact match and 78-100% for an adjacent match for the M2F dataset (Table 3).

For the M2F dataset there was little-to-no difference between the soil map accuracies calculated at point-support (M2F-PS) relative to accuracies calculated at field-support (M2F-FS) (Table 3). The average farm size across the 75 farms was 2.4 ha (SD ± 2.0) and our delineation procedure captured, on average, 48% of a field's area, with a range of 8% to 100%. On average these delineated areas intersected 1.8 SoilGrids pixels (range: 1-4 pixels) and 8.6 iSDA pixels (range: 2-24). Due to the large size of HWSD and WISE map unit polygons in Ghana, all farms fell within a single map unit and thus were attributed with the dominant component property value for both the point-support and field-support cases.

When we expanded our measure of prediction accuracy to include adjacent classes (i.e., $OA_{adj}$), classification accuracy increased to 39-85% for soil texture and 73-93% for rock fragment volume (Table 3). Individual soil texture class and rock fragment class producer accuracies for the FTF-M2F-PS and M2F-PS datasets are show in Figures 6 and 7, respectively.

Although 51% of LandPKS site-based texture measurements were either sand or loamy sand, none of the web-based soil maps predicted these classes at any of the sampling sites and at only <1% across all of Ghana (Fig. 4). Soil texture classes with higher prediction accuracies included sandy clay loam, sandy loam, loam, and clay loam, which corresponded to the most common texture classes predicted among the four maps (Figs. 4,6,7). A similar trend occurred for rock fragment volume class (Figs. 4,6,7).

Table 3. Accuracy of soil map predictions for texture class and rock fragment volume class from the three evaluation datasets

| Soil Texture Class | | | | |
|---|---|---|---|---|
| | **HWSD** | **WISE** | **SoilGrids** | **iSDA** |
| **FTF-M2F-PS** | | | | |
| OA | 0.09 | 0.14 | 0.08 | *0.39†* |
| $OA_{adj}$ | 0.40 | 0.49 | 0.39 | *0.85†* |
| BER | 0.91 | 0.91 | 0.90 | *0.76†* |
| OA-P‡ | 0.41 – 0.55 | 0.46 – 0.64 | 0.38 – 0.54 | *0.71 – 0.91†* |
| $OA_{adj}$-P ‡ | 0.45 – 0.55 | 0.54 – 0.64 | 0.44 – 0.54 | *0.90 – 1.00†* |
| **M2F-PS** | | | | |
| OA | 0.27 | 0.16 | 0.33 | 0.28 |
| $OA_{adj}$ | 0.74 | 0.82 | 0.81 | 0.81 |
| BER | 0.90 | 0.95 | 0.80 | 0.88 |
| OA-P‡ | 0.63 – 0.85 | 0.52 – 0.74 | 0.69 – 0.91 | 0.64 – 0.86 |
| $OA_{adj}$-P ‡ | 0.78 – 0.99 | 0.86 – 1.00 | 0.85 – 1.00 | 0.85 – 1.00 |
| **M2F-FS** | | | | |
| OA | 0.28 | 0.15 | 0.35 | 0.32 |
| $OA_{adj}$ | 0.80 | 0.85 | 0.84 | 0.90 |
| BER | 0.87 | 0.94 | 0.75 | 0.84 |
| OA-P‡ | 0.64 – 0.86 | 0.51 – 0.73 | 0.71 – 0.93 | 0.68 – 0.90 |
| $OA_{adj}$-P‡ | 0.84 – 1.00 | 0.89 – 1.00 | 0.88 – 1.00 | 0.94 – 1.00 |
| **Rock Fragment Volume Class** | | | | |
| | **HWSD** | **WISE** | **SoilGrids** | **iSDA** |
| **FTF-M2F -PS** | | | | |
| OA | 0.27 | 0.29 | 0.26 | *0.33†* |
| $OA_{adj}$ | 0.72 | 0.73 | 0.73 | *0.91†* |
| BER | 0.83 | 0.81 | 0.83 | *0.78†* |
| **M2F-PS** | | | | |
| OA | 0.59 | 0.59 | 0.33 | 0.59 |
| $OA_{adj}$ | 0.87 | 0.87 | 0.88 | 0.87 |
| BER | 0.80 | 0.80 | 0.83 | 0.79 |
| **M2F-FS** | | | | |
| OA | 0.56 | 0.56 | 0.37 | 0.56 |
| $OA_{adj}$ | 0.89 | 0.89 | 0.93 | 0.89 |
| BER | 0.80 | 0.80 | 0.82 | 0.80 |

†iSDA accuracy statistics for the 'FTF-M2F-PS' dataset are not reliable due to the partial use of this dataset (i.e., LandPKS Feed the Future sites) in iSDA model training/validation.

OA, overall accuracy; OA$_{adj}$, overall accuracy when accounting for class adjacency to the correct property class; BER, balanced error rate; FTF-M2F-PS, dataset containing all LandPKS sites in Ghana at point-support (6,514 sites, 19,542 soil layers); M2F-PS, LandPKS Map to the Future dataset at point-support (225 sites; 675 soil layers); M2F-FS, LandPKS Map to the Future dataset at field-support (75 sites; 225 soil layers).

  ‡Potential overall accuracies (OA-P) and potential overall class adjacent accuracies (OA$_{ad}$-P$_j$) accounting for
potential uncertainty in the evaluation data. The reported range represents the potential increase in OA$_{adj}$ due to potential error in evaluation class labels.

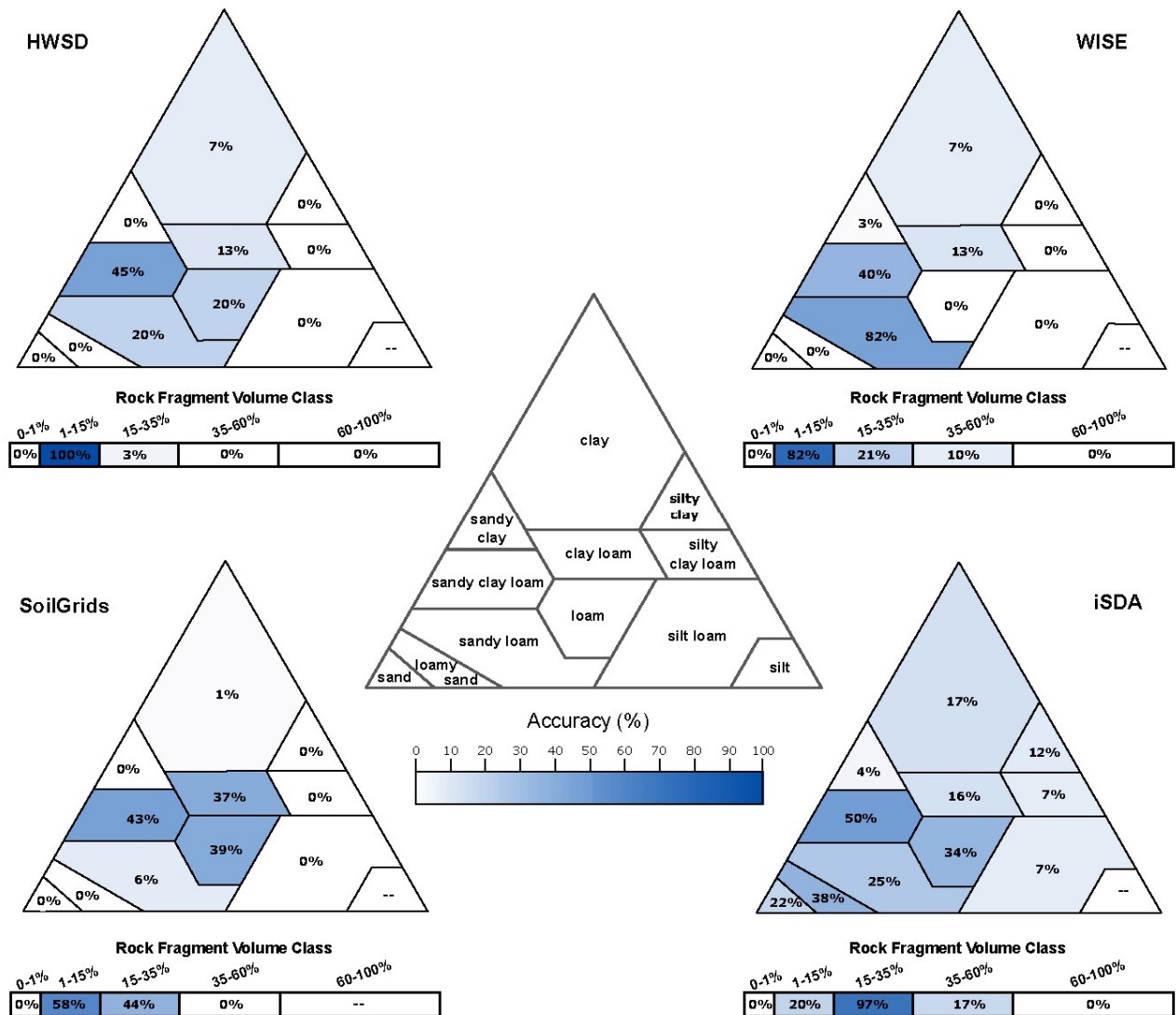

**Figure 6. Soil texture class and soil rock fragment volume class Producer's accuracy for the four soil maps based on the FTF-M2F dataset.**

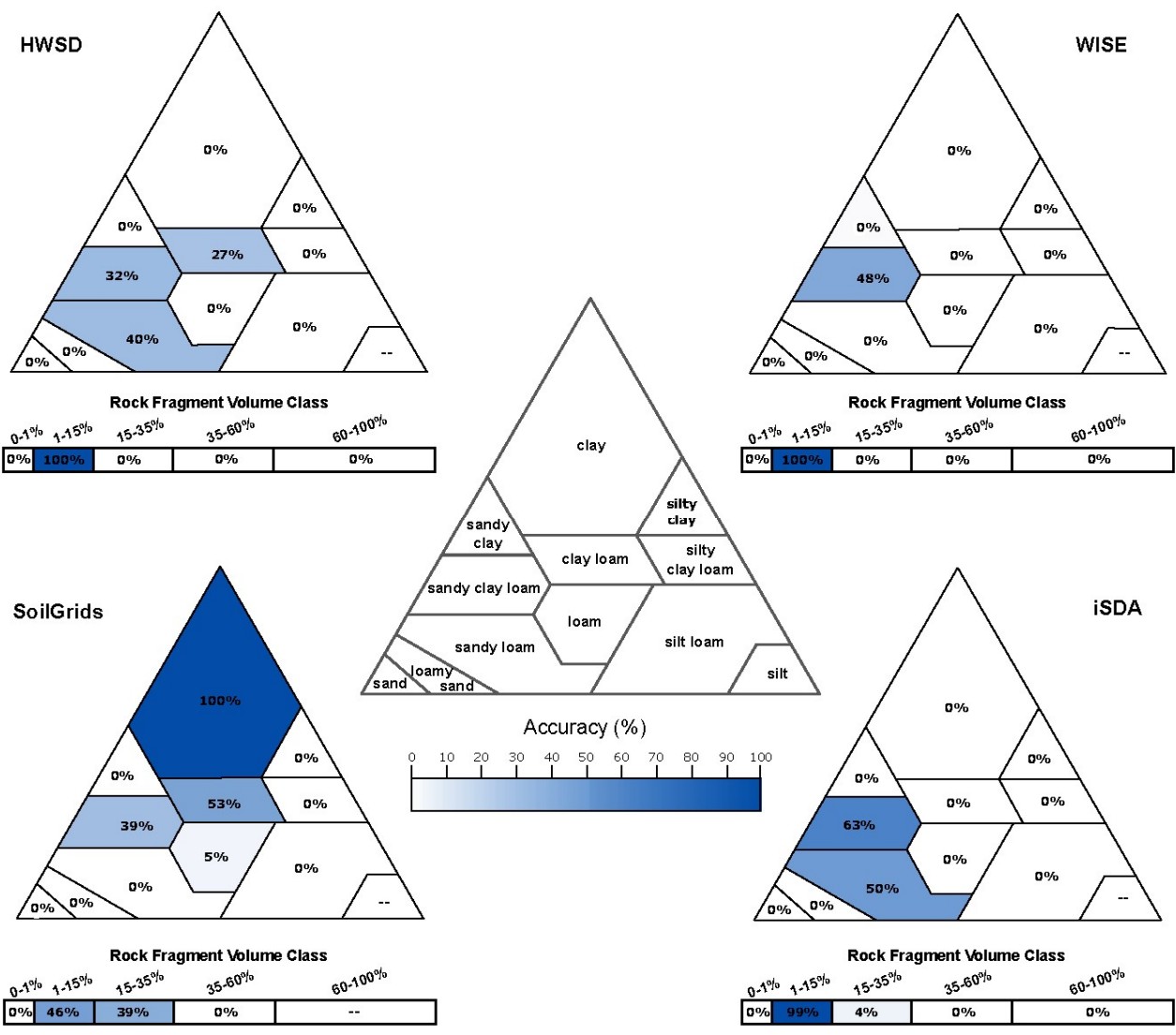

**Figure 7. Soil texture class and soil rock fragment volume class Producer's accuracy for the four soil maps based on the M2F dataset.**

### 3.2.2 Agro-Ecological Zone Soil Suitability

The distribution of maize soil suitability ratings and classes calculated using LandPKS site-base data and soil map data are shown in Figure 8. Soil suitability ratings were noticeably different between LandPKS and the soil maps, with LandPKS values being substantially lower than those of the soil maps. Furthermore, the range of suitability ratings for LandPKS was significantly wider than that of the soil maps for both datasets (Fig. 8a,b). While iSDA appears to capture these lower suitability ratings (Fig. 8a), when we look at the independent test dataset (M2F) we see that iSDA fails to detect these lower suitability soils (Fig. 8b). When suitability ratings are translated to suitability classes, these differences are further emphasized, with sampling sites classified in the top two suitability classes for the soil maps, whereas sampling sites for LandPKS were more evenly distributed across the suitability classes (Fig. 8c,d). While both the predictive and conventional soil maps were classified in either the 'No constraint' or 'Slight constraint' classes, 65% of sampling sites (17% of M2F sites) were classified as having moderate to very severe soil constraints based on LandPKS site-specific data (Fig. 8c,d). OA for the GAEZ suitability classes were similar for all four soil maps, at just 15-18% for the complete dataset and 27-61% for the M2F datasets (Table 4). High PA for the 'No constraint' suitability class and low-to-zero percent accuracies for the other suitability classes further show the over-prediction of the 'No constraint' suitability class among the four maps (Table 4). A PA of zero for a suitability class means that the maps did not correctly predict any of the field observations of that class. UAs that fail to return a value indicate that the map failed to predict any values of that class (Table 4).

Analysis of the individual soil quality indices revealed that most sites were limited by their availability of soil nutrients, with 50% of LandPKS sites having a moderate to very severe constraint (Fig. 9a). Far fewer sites were constrained by rooting conditions or workability, with only 15% and 2% of LandPKS sites having a moderate to very severe constraint for rooting conditions (Fig. 9b) and workability (Fig. 9c), respectively. Nutrient availability was a main source of limitations identified for HWSD but not for either WISE or SoilGrids. HWSD and WISE also had limitations identified for soil rooting conditions in a small subset of sites. No limitations were identified for workability by any of the soil maps (Fig. 9c).

**Table 4. Accuracy of soil map predictions for AEZ soil suitability classes**

| Suitability classes | HWSD | | WISE | | SoilGrids | | iSDA | |
|---|---|---|---|---|---|---|---|---|
| | PA | UA | PA | UA | PA | UA | PA | UA |
| **FTF-M2F -PS** | | | | | | | | |
| No constraint | 0.69 | 0.14 | 0.83 | 0.18 | 0.94 | 0.15 | *0.84†* | *0.43†* |
| Slight constraint | 0.21 | 0.17 | 0.26 | 0.18 | 0.02 | 0.28 | *0.40†* | *0.27†* |
| Moderate constraint | 0.00 | -- | 0.00 | -- | 0.00 | 0.00 | *0.43†* | *0.45†* |
| Severe constraint | 0.00 | -- | 0.00 | -- | 0.00 | -- | *0.16†* | *0.34†* |
| Very severe constraint | 0.00 | -- | 0.00 | -- | 0.00 | -- | *0.09†* | *0.92†* |
| Not suitable | 0.00 | -- | 0.00 | -- | 0.00 | -- | *0.00†* | -- |
| **OA** | **0.15** | -- | **0.18** | -- | **0.15** | -- | *0.39†* | -- |
| **M2F-PS** | | | | | | | | |
| No constraint | 0.39 | 0.40 | 1.00 | 0.56 | 0.52 | 0.58 | 1.00 | 0.56 |
| Slight constraint | 0.19 | 0.11 | 0.00 | -- | 0.52 | 0.28 | 0.00 | -- |
| Moderate constraint | 0.00 | -- | 0.00 | -- | 0.00 | -- | 0.00 | -- |
| Severe constraint | 0.00 | -- | 0.00 | -- | 0.00 | -- | 0.00 | -- |
| Very severe constraint | 0.00 | -- | 0.00 | -- | 0.00 | -- | 0.00 | -- |
| Not suitable | -- | -- | -- | -- | -- | -- | -- | -- |
| OA | **0.27** | -- | **0.56** | -- | **0.43** | -- | **0.56** | -- |
| **M2F-FS** | | | | | | | | |
| No constraint | 0.39 | 0.44 | 1.00 | 0.61 | 0.50 | 0.61 | 1.00 | 0.61 |
| Slight constraint | 0.21 | 0.12 | 0.00 | -- | 0.53 | 0.27 | 0.00 | -- |
| Moderate constraint | 0.00 | -- | 0.00 | -- | 0.00 | -- | 0.00 | -- |
| Severe constraint | 0.00 | -- | 0.00 | -- | 0.00 | -- | 0.00 | -- |
| Very severe constraint | -- | -- | -- | -- | -- | -- | -- | -- |
| Not suitable | -- | -- | -- | -- | -- | -- | -- | -- |
| OA | **0.29** | -- | **0.61** | -- | **0.44** | -- | **0.61** | -- |

†iSDA accuracy statistics for the 'FTF-M2F-PS' dataset are not reliable due to the partial use (LandPKS Feed the Future dataset) of this dataset in iSDA model training/validation.

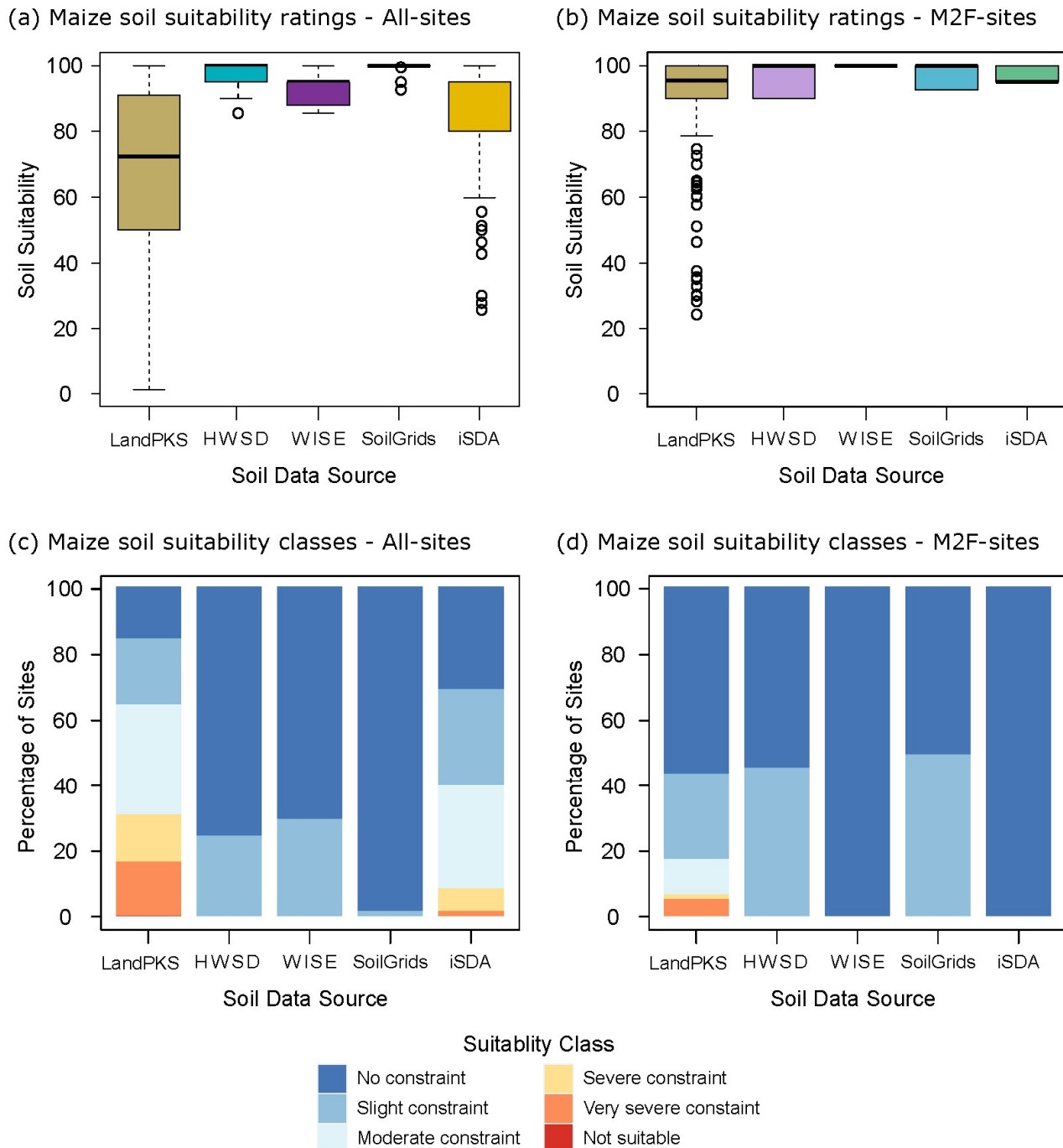


**Figure 8. Distribution of GAEZ soil suitability ratings (a,b) and classes (c,d) for the FTF-M2F and M2F soil sampling sites based on the five different soil data sources.**

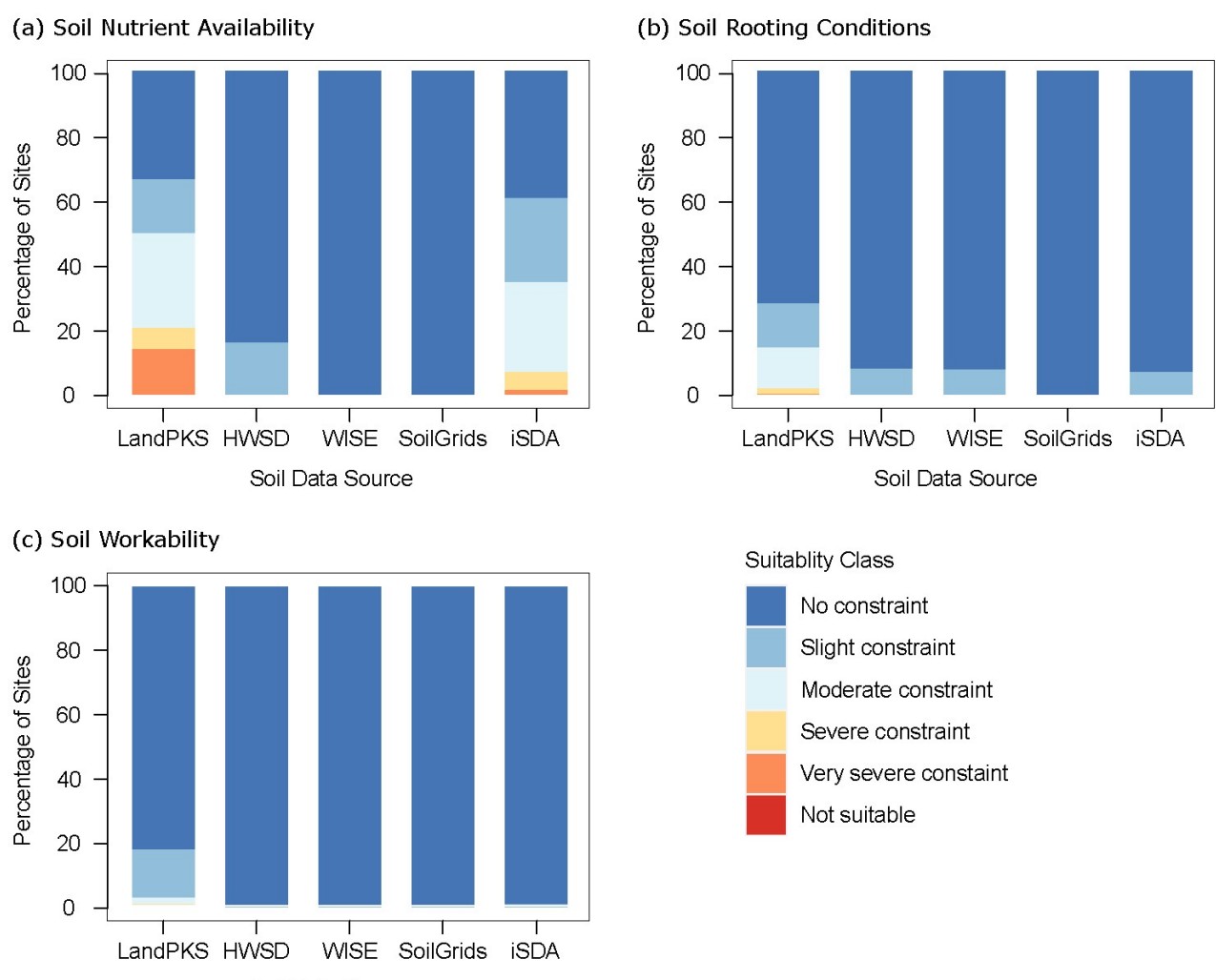

**Figure 9. Distribution of GAEZ soil quality index classes for (a) soil nutrient availability (b) soil rooting conditions, and (c) soil workability for the FTF-M2 dataset (6,514 soil sampling sites) based on the five different soil data sources.**



## 4. Discussion

### 4.1 Evaluation of soil map accuracy

The utility of a soil map depends on its intended use and the level of accuracy required for that use. When applied at a regional scale, current soil maps have been used effectively to inform agronomic and environmental policies. However, less is known about the accuracy of soil maps at the farm/field scale and whether soil map data at this scale is sufficiently accurate to inform site-specific land management. Differences in soil properties between sites, like pH or texture, can result in highly different

management requirements. For example, many essential plant nutrients become increasingly unavailable in soils at low pH (e.g., <4.5), making any efforts to fertilize acidic soils ineffective. Acidic soils require the application of amendments (e.g., lime) to raise the soil pH before any inherent nutrient deficiencies can be addressed. Accurately identifying these site-specific soil differences is critical for addressing the soil limitations that currently inhibit crop yields. Results from this study show that publicly available web-based soil maps of Ghana lack the needed accuracy to reliably inform soil management decisions on

smallholder farms (i.e., 1-2 ha). Standard measures of map accuracy for the class-based soil properties (i.e., texture class, rock fragment class) showed that all the soil maps were equally inaccurate in estimating the correct property class, predicting the wrong texture class 6-9 times out of 10 and the wrong rock fragment class 4-7 times out of 10. A similar study in Namibia evaluated the accuracy of surface soil texture estimates from seven soil maps (including HWSD, WISE and SoilGrids) (Buenemann et al., 2021).  This study found that soil maps in Namibia predicted the correct topsoil texture class in only 13%

to 42% of test sites, indicating that none of the maps were sufficiently accurate for most site-specific management applications. Another study in Rwanda evaluated the accuracy of SOC and pH predictions from AfsoilGrids250 maps (Söderström et al., 2017). Söderström et al. (2017) found that the AfsoilGrids250 soil map predictions in Rwanda were poorly correlated to an independent validation dataset, with coefficients of determination of 0.05 and 0.11 for SOC and pH, respectively.

Accuracy assessments based on exact matching of soil classes, however, can underrepresent the functional accuracy of

soil properties. For example, two soils with the same clay content (15%) but slightly different sand contents (51 vs 53% sand) would fall into two different soil texture classes (sandy loam and loam, respectively) due to their proximity to the texture class boundary (Fig. 10a). If we predicted both soils to be sandy loam, our accuracy would only be 50% even though both soils may function like a sandy loam. Accounting for class adjacency in the overall accuracy evaluation accounts for these 'near misses', providing a less restrictive assessment of map accuracy. However, although class-adjacent accuracies in this study were higher

than overall accuracies for texture across all soil maps, they only increased to 39-49% for the FTF-M2F dataset, indicating that 50-60% of map-based soil texture estimates were considerably different (i.e., greater than one texture class difference) than site-based estimates.  Furthermore, even after accounting for potential inaccuracies in our evaluation dataset, texture class accuracies only increased to 38-64% FTF-M2F dataset. A similar result was found in Namibia, where topsoil texture predictions were often more than one textural class away from the site-based classes (Buenemann et al., 2021). In the smaller

M2F dataset, class-adjacent soil texture accuracies were considerably higher ($OA_{adj}$: 74-90%; $OA_{adj}$-P: 78-100%) at both point-

and field-support, most likely due to the higher proportion of finer textured soils which were more accurately predicted by the soil maps in these areas. (Table 3, Fig. 1).

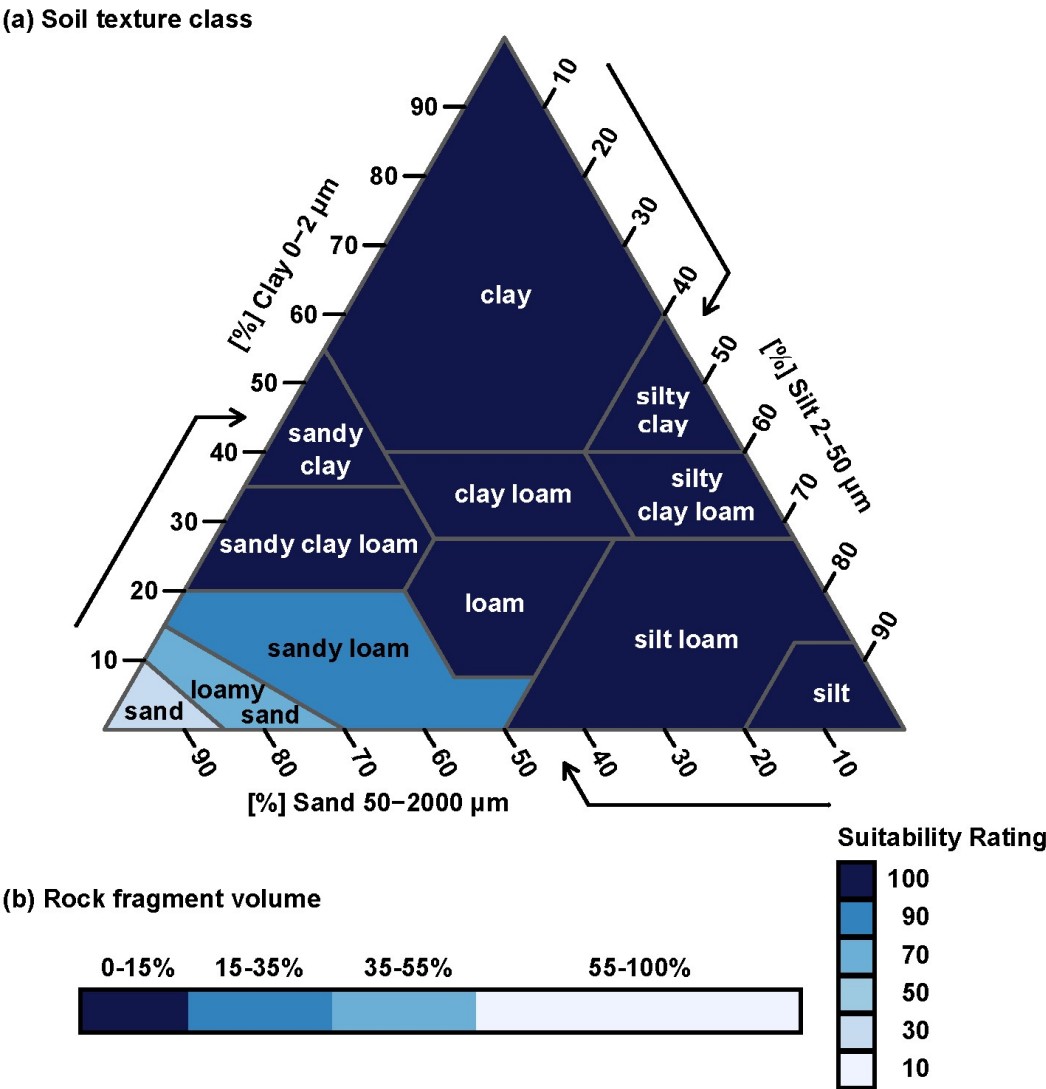

**Figure 10. GAEZ soil nutrient availability ratings for the different soil texture and rock fragment volume classes based on a rainfed, low-input maize production system.**


While comparing relative differences in soil property values can provide insight into the accuracy of map-based estimates, it is often difficult to interpret the functional implications of those differences. Modelling frameworks like GAEZ provide a way to translate soil property differences into crop-specific soil quality indices and soil suitability ratings that can compare soil

functional differences. Additionally, the GAEZ framework provides a more holistic means of comparing soils since each soil suitability rating is calculated based on all soil properties and across all soil depths at a site, providing a single functional measure at each location. This contrasts with standard measures of map accuracy that are based on a single soil property at only one soil depth.

In our application of the GAEZ framework (low-input, rain-fed maize) the soil property rating criteria were based on two different levels of soil property generalization. The first based on the groupings of soil property values into soil property classes (Figs. 6,7) and second, the broad grouping of soil classes into soil suitability ratings (e.g., Fig. 10). This means that in many cases soil property differences resulting from either measurement or prediction error will be minimized due to these large within-group property ranges. This is not the case, however, within certain regions of the soil property space. For example, in the GAEZ system, soil texture poses no limitations to nutrient availability for all texture classes except the three classes with the highest sand content (sandy loam, loamy sand, sand) which pose increasing limitations with increasing sand content (Fig. 10). Thus, to accurately assess this limitation one must accurately differentiate a sand, loamy sand, or sandy loam from any texture finer than a sandy loam. On the other end of the soil textural triangle, the clay texture class negatively impacts the soil quality ratings for rooting condition and workability, while all other texture classes do not pose any limitations. Despite these relatively narrow soil property ranges for identifying crop constraints, 65% of sites were classified as having moderate to very severe soil constraints, while the soil maps all failed to predict these high constraint classes at any of the study sites. The low functional accuracy of soil maps in Ghana based on our modified GAEZ framework was due to several contributing factors: (1) constraints on maize soil suitability were largely confined to coarse textured soils (i.e., sand, loamy sand, sandy loam), (2) 71% of site-based texture estimates were in coarse texture classes, and (3) soil maps had low prediction accuracy for the coarse texture classes. In areas dominated by medium-to-fine textured soils, functional accuracies would likely be much higher due to the wide range of texture classes (e.g., sandy clay loam vs silty clay) that are rated functionally similar (Fig. 10a).

### 4.2 Estimating site-based soil properties: Potential sources of error

### 4.2.1 Sources of field sampling error

When evaluating the accuracy of soil map predictions, two sources of error can occur from the field sampling protocol: the first originating from the sampling design and second from the sampling methodology. Several recent studies have shown that unbiased assessments of soil map accuracy require independent test datasets that have been generated using probability-based sampling designs (Brus et al., 2011; Piikki et al., 2021). The datasets used in this study were purposive, focusing on smallholder farmers in Ghana, and spatially clustered in northern Ghana, with only a small subset of farms in southern Ghana. However, within this specific land use type, probability-based sampling methods were used to select sites from a larger population of smallholder farms. Therefore this assessment should only be viewed in the context of utilizing soil maps to help inform smallholder farmers, and not their utility for informing other land use types (e,g., forestry, grazing lands) which may provide more accurate soil predictions.

Field sampling error can also occur due to differences in sampling methodology. This study used field-estimated soil property values as reference data for evaluating the accuracy of soil map predictions. However, soil map predictions of texture (i.e., sand, silt and clay mass fractions) are typically made using laboratory measurements for input data. While field estimation of soil texture using simple dichotomous keys has been shown to produce relatively accurate estimates when compared to laboratory measurements (Minasny et al., 2007; Richer-de-Forges et al., 2022; Salley et al., 2018; Vos et al., 2016b), an explicit quantification of field texture estimation uncertainty is needed to assess the feasibility of using field texture data to evaluate soil map predictions. Our global meta-analysis of field texture estimation uncertainty revealed generally good agreement between field and laboratory measurements for very coarse and very fine texture classes (e.g., sand and clay) and lower agreement for medium texture classes (e.g., silty clay loam and loam) (Table 2). The overall accuracy of our soil texture class evaluation datasets was moderate (OA: 42-68%) but increased substantially when accounting for adjacent texture class matches (OA$_{adj}$: 75-97%). This higher uncertainty in estimating an exact match (i.e., field texture = lab texture) made it difficult to estimate realistic ranges for the potential accuracy of soil map predictions. However, the lower uncertainty in the adjacent class match allowed us to estimate a more constrained and thus likely more realistic range of potential map accuracy when matching to adjacent classes (OA$_{adj}$-P). Our higher confidence in the estimated range of OA$_{adj}$-P and the similarity in estimated accuracy ranges for OA-P and OA$_{adj}$-P, suggests that the actual range for OA-P is likely lower than what was estimated based on the estimated uncertainty of our evaluation datasets.

In evaluating the utility of field-based evaluation datasets, it is important to recognize that laboratory measurements have their own inherent sources of error. For example, in the 6[th] FSCC interlaboratory comparison it was reported that clay content was one of the most difficult properties to consistently measure, with a coefficient of variation (CV) of 32% among 50 participating laboratories (Cools and Vos, 2010). Additionally, a recent interlaboratory comparison among three soil laboratories in Ghana revealed significant variation in soil texture measurements, with a CV of 47% for clay based on 10 soils with contrasting textures (2020, unpublished data). Such high interlaboratory variability may result from the fact that some soils, due to their minerology or chemical make-up, are not well-suited for laboratory particle-size analysis; in which case field-based estimates may be considered more reliable than lab data (Landon, 1988). This is true for highly weathered oxide-rich tropical soils where traditional laboratory techniques often underestimate clay content due to the soil's resistance to dispersion (Silva et al., 2015). These types of highly weathered and/or oxide rich soils (Plinthisols, Ferralsols, Acrisols) are common throughout our study area. It is therefore important to be cognizant of the potential uncertainties associated with laboratory soil texture measurements, especially in tropical regions, and the implication of these uncertainties on the use of field texture estimates.

When evaluating soil data uncertainty, it is important to determine the level of measurement precision needed to inform a particular outcome. For example, while soil particle size mass fractions (high measurement precision) are often required for soil modelling, soil texture classes (lower measurement precision) are generally sufficient for on-farm soil management. Lowering the level of measurement precision (e.g., using soil texture classes in place of particle size fractions) also minimizes the uncertainty associated with different data sources (e.g., laboratory, field). Functional soil assessments often require even

lower levels of measurement precision as demonstrated by the GAEZ framework where the soil suitability ratings and classes have relatively low measurement precision (Fig. 10), which in turn further minimized the inherent uncertainty associated with both lab and field-based measurements. Furthermore, in our evaluation of functional accuracy using the modified GAEZ framework, functional differences for soil texture only occur for textural classes either high in sand (sand, loamy sand, sandy loam) ($SQ_N$, Fig. 10) or high in clay (clay) ($SQ_W$). The generally higher accuracy of hand texture estimates in these regions of soil texture space decreases the probability of sampling error in our functional accuracy evaluation.

### 4.2.2 Sources of map error: spatial uncertainty

Soil maps are created at defined spatial scales, producing map information (e.g., field data, assigned classes, spatial delineations, interpretations) that is constrained by the patterns and characteristics of those scales (Soil Survey Division Staff, 2017). Both conventional and predictive soil maps account for spatial uncertainty in different ways. For conventional soil maps, the mapping scale determines the size and purity of soil map units, where small map scales (e.g., 1:5,000,000) contain large map units comprised of multiple soil components, while large map scales (e.g., 1:12,000) contain smaller map units that are often comprised of a single soil component. The small map scale of HWSD and WISE (1:5,000,000) resulted in individual map unit polygons ranging in area from 61 $km^2$ to 17,947 $km^2$. Across these vast areas each map unit is only attributed with a few soil components whose spatial delineation within each polygon is unspecified. The most common approach to deal with this spatial uncertainty is to attribute each polygon to its dominant soil component, as was done in this study. Depending on the number of soil components in a map unit and their areal extents, it is possible for the dominant component to comprise only a small percentage (e.g., 20%) of a large map unit area. Given the large spatial extent of the map unit polygons in the study area and our generalization of map units based on dominant component, it is not surprising that these map products resulted in low site-specific accuracies.

Predictive soil maps are faced with a different set of challenges relating to spatial uncertainty. Since predictive soil maps use raster based environmental data as their predictive covariates, the spatial resolution of covariate data determines the spatial scale of the resulting soil map, which imparts an implied level of precision to the end-user. The accuracy of predictive soil maps, however, depends on the characteristics of both the soil point data and covariate data used to build the models. An important characteristic of the soil point data is how well it represents the variability of covariate data across the entire inference space. Predictive soil maps often use existing field data from soil surveys that were conducted at different spatial scales to train and validate their models, and therefore may not adequately represent the full covariate information space. This can result in cases where the global model accuracy is high but local model accuracies are low, because certain geographic regions within the prediction area are poorly represented in covariate space. This was seen in the case of iSDA where global prediction accuracies (Concordance Correlation Coefficient) for sand, silt, and clay ranged from 0.78-0.85 (Hengl et al., 2021), yet texture class prediction accuracies in Ghana were low (OA: 0.28-0.32). Similarly, SoilGrids global prediction accuracies (model efficiency coefficient) for sand, silt, and clay ranged from 0.40-0.70 (Poggio et al., 2021), while SoilGrids texture class

prediction accuracies in Ghana were low (OA: 0.08-0.35). While these map products were not able to provide sufficiently accurate soil property predictions for site-specific management within the study area, high global prediction accuracies for many of the modelled soil properties indicates that these maps have higher accuracies across larger spatial/variance scales.

In predictive soil modelling, model error is composed of two components: bias, which relates to model accuracy; and variance, which relates to model precision or uncertainty. SoilGrids and iSDA maps both employ ensemble modelling approaches to calculate spatial predictions of model uncertainty (SoilGrids: Quantile Random Forest; iSDA: Ensemble Bootstrapping). Ensemble models are effective at increasing model accuracy and are often implemented using some form of model averaging (Polikar, 2012). Assuming the different soil models produce different errors at each location, averaging the model outputs generally reduces the error (model bias) by averaging out the error components. A downside of model averaging is that it has a smoothing (variance-reducing) effect which can remove valid information from the outer ranges of the soil property distribution. There are many cases where the ability to predict these 'extreme' values is crucial, for example, at the smallholder farm scale where the risk or cost associated with incorrectly identifying soil constraints can be high for cash-constrained farmers. Depending on the soil management scenario, there are financial costs associated with both false negative results (i.e., failure to detect constraint – Type 2 error; e.g., failure to lime a very strongly acidic soil before applying fertilizer) and false positive results (i.e., false detection of constraint – Type 1 error; e.g., applying lime to a neutral soil). While the mean or median predicted soil values may not indicate the presence of soil constraints, spatial predictions of model uncertainty can be used to determine where constraints have a predicted probability of occurrence. Future research is needed to evaluate information on soil map uncertainty and how this information can be effectively communicated and incorporated into smallholder agronomic decision making.

The increasing availability of higher spatial resolution environmental covariates has led to expanded efforts to produce finer spatial resolution soil predictions. However, the relationship between each soil property or class and the covariate data can be scale-dependent, meaning that the spatial scale (i.e., grid resolution and spatial extent) at which a covariate is calculated can affect the strength of its relationship to the modelled property. Thus, higher spatial resolution covariates do not always translate to more accurate fine-scale model predictions, and in some cases model accuracy may decrease due to the scale-dependency of the predictor-covariate relationships. Several studies have demonstrated these spatial scaling effects for terrain attributes, where the highest model accuracies did not correspond to the terrain attributes calculated at the finest spatial scales (Kim and Zheng, 2011; Maynard and Johnson, 2014; Roecker and Thompson, 2010).

Growing recognition of the need for site-specific soil data has prompted efforts to produce finer spatial resolution soil data from existing soil maps. For example, disaggregation techniques are being used to delineate the location of soil map unit components within conventional soil maps (Häring et al., 2012; Nauman and Thompson, 2014; Vincent et al., 2018) and predictive soil maps are using higher spatial resolution covariates to make higher spatial resolution predictions (e.g., iSDA). However, it is important to recognize that the primary data (e.g., map unit polygons, point data), metadata, and inherent decisions made at the original soil mapping scale remain determinate, where those original biases persist across scales. These initial biases can be compensated for through targeted sampling to expand and refine the model inference space (Soil Survey

Division Staff, 2017). For example, Stumpf et al. (2017) used model uncertainty to guide additional sampling efforts for model refinement, while other studies have used additional sampling to refine regional-to-continental scale soil map predictions within a localized area for farm-scale applications (Piikki et al., 2017; Söderström et al., 2017).

### 4.2.3. Sources of map error: temporal uncertainty

Current soil maps provide models of soil spatial variation that ignore temporal changes in soil properties. Conventional soil maps are created over years-to-decades and populated with soil data that represents the modal concept of soil types. Predictive soil maps use soil profile datasets collected over multiple decades which are correlated to environmental covariates that often represent either a single point in time or some aggregate value calculated from a fixed time interval. If significant soil degradation occurs sometime after the soil profile data was collected, the covariate data at that site may no longer correspond to the original soil property values. This can weaken or introduce confusion into the modelled relationship between soil property data and environmental covariates, which in turn can negatively impact the accuracy of model predictions (Owusu et al., 2020). Our evaluation of soil map accuracy in this study focused on static soil properties, which in theory should provide more accurate map estimates relative to measures of soil health (e.g., soil nutrients) that change in response to land use and management over short time scales (i.e., years-to-decades). However, in high erosional or depositional environments these static properties can also change over relatively short time scales, which may have contributed to the low map accuracy for these static soil properties in this study. Furthermore, these results suggests that efforts to map more dynamic properties using either conventional or predictive mapping approaches would likely produce estimates with even greater uncertainty.

### 4.3. Implications for site-specific soil management

To close existing yield gaps smallholder farmers must identify the factors that constrain productivity. Many yield limiting factors are directly or indirectly soil-related, including nutrient deficiencies, susceptibility to drought, soil compaction, waterlogging, high erosion risk, etc. Obtaining accurate site-specific soil data is a first step towards uncovering soil-based limitations and implementing management practices that can mitigate these production constraints. Current web-based soil maps of Ghana fail to meet the accuracy requirements for site-specific farm management or even for farm-level land use planning. Field-based texture assessments like the ones used in this study, coupled with ongoing advancements in soil mapping and on-site verification technologies like proximal sensors (Piikki et al., 2016; Viscarra Rossel et al., 2011) and smartphone-based decision support tools (Herrick et al., 2013; O'Geen et al., 2017), can help constrain the uncertainty associated with site-based soil map predictions.

There is a need for improved technologies that can assist farmers in identifying their soil characteristics, and matching those characteristics to appropriate inputs and technologies that can enhance the long-term production capacity of their soils (Berkhout et al., 2015). A useful conceptual model employed by conventional soil maps is the grouping of soils into soil types based on both field-described morphology data and laboratory analysis. Soil types convey information on the general range of soil behavior a land manager can expect in response to management actions and disturbance effects. Through identifying the

soil type at a location, smallholder farmers can gain a better understanding of potential soil limitations and the most appropriate management strategies for improving soil health and crop yields. The concept for each soil type is based on a set of reference soils which define the representative soil property distributions for each soil type. Soil types that have been intensively managed over long periods of time, however, can deviate significantly from these representative property ranges. Thus, in addition to understanding the soil type, information on a site's management history and current resource allocation are needed

to better assess general soil health and possible soil-related limitations.

       To make soil information actionable for smallholder farmers, soil information needs to be contextualized for their intended land use. For example, a maize farmer needs to know how their soil texture, rock fragment content, and soil depth will affect crop growth. Based on their soil type and management history, farmers also need to know what type of fertilizer to apply, in what amount, and when to apply it for optimal crop uptake. This study demonstrated how downscaling the GAEZ soil

suitability framework provides a way to interpret site-specific soil information for crop-specific soil management. While this study applied a modified version of GAEZ based on static soil properties, this approach could also incorporate dynamic soil properties that influence soil nutrient availability as well as other soil quality indices.

## 5. Conclusions

Many agronomic constraints are directly or indirectly soil-related, and therefore accurate site-specific soil information is

needed to address these limitations. Technological advancements are facilitating the creation of soil maps at high spatial resolutions which impart an implied level of precision. The accuracy, and thus utility, of these maps for applications at large spatial scales is often unknown. This study evaluated both the relative and functional accuracy of four publicly available web-based soil maps of Ghana and found that in most cases these map products are not accurate enough to inform site-specific soil management. We found that overall accuracies for soil texture and rock fragments predictions ranged from 8-14% and 26-

29%, respectively. However, when accounting for potential uncertainty in the evaluation dataset, soil texture class accuracies could be as high as 38-64%. Overall accuracies that allow for class adjacency increased for both soil texture (39-49% or 44-64% after adjusting for evaluation uncertainty) and for rock fragments (72-73%). Traditional measures of map accuracy, however, can be misleading since small differences in soil property values, while technically different, may be functionally similar. To account for this, we used a modified version of the GAEZ soil suitability framework to evaluate the functional

accuracy of the soil map predictions. This functional assessment confirmed the results from the standard accuracy assessment, with overall accuracies for soil suitability classes ranging from 15-61%. Results from this study highlight the variable site-specific accuracy of current soil map information and the potential implications for on-farm decision making. The urgent need for reliable soil information, that is, information with a specified accuracy and precision for a targeted objective (e.g., attainable crop yield), has become increasing clear and many areas of research are being advanced to address this global challenge.

Among these is the continued improvement of soil maps, particularly through the advancement of predictive soil mapping technologies, including improved predictive algorithms, expanded soil training/testing datasets, and advancements in the

quantification of model uncertainty. For example, both predictive soil maps evaluated in this study (SoilGrids, iSDA) provide uncertainty maps and future work is needed to utilize this information for large-scale site-specific analysis. There is also a need for the training of more soil scientists to expand the characterization and sampling of soil landscapes with high information uncertainty resulting from a lack of ground-truthed samples and/or poorly understood soil landscape relationships. Soil scientists can also assist in the training of non-soil specialists (e.g., field enumerators, citizen scientists), who, with the help of on-site verification technologies like smartphone-based decision support tools (e.g., LandPKS) and proximal sensors (e.g., VisNIR), can collect high quality field-based soil data. This information can then be used both directly to inform site-specific decision making (e.g., smallholder fertilizer application rates), as well as to improve soil map predictions, as demonstrated by the iSDA soil map which used LandPKS field data to generate model predictions. All these advancements can help constrain the uncertainty associated with site-based soil map predictions and help provide access to accurate soil property information so urgently needed by smallholder farmers to improve soil health and enhance the long-term production capacity of their soils.

**Code and Data availability.**

All data used in this study are available from public data repositories. All R code required to reproduce the content of this study will be made publicly available on Github.

**Author contributions.**

JM, JH, and JN conceived and designed the study. JH and JN secured funding. EY and SO performed field work. JM performed the analysis, visualization, and drafted the manuscript. All authors (JM, EY, SO, MB, JN, JH) contributed to the review, interpretation, and writing of the final manuscript.

**Competing interests.**

The authors declare that they have no conflict of interest.

**Acknowledgements.**

We gratefully acknowledge the valuable inputs and cooperation from the Grameen Foundation.

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
