# Peer review of "Accuracy of regional-to-global soil maps for on-farm decision making: Are soil maps "good enough"?"

_EGUsphere, 2022_

## Author Response (AR1)

Dear Editor,

Thank you for the opportunity to revise our manuscript. We have addressed all comments from both reviewers. Please find line numbers corresponding to the revised manuscript for all major revisions.

Kind regards,

Jonathan

**Response to review by Dr. Colby Brungard (RC1)**

**Reviewer Comment:**
*"I have two concerns with this analysis. My main concern with the research regards the (im)precision of the validation data. The methodology that was used to evaluate the soil map accuracy assumes that there is no uncertainty in the labels of the validation observations (e.g., soil texture or rock fragment classes), even though the authors recognized the error/uncertainty in field collected observation in section "4.2.1 Sources of field sampling error". I am uncomfortable with this because I feel that not accounting for measurement error in the validation data has the potential to obfuscate the 'true' accuracy of the soil maps (either higher or lower than is presented)."*

**Author Response:**
While Dr. Brungard makes a valid point here, we feel that this issue is not confined to field collected validation data. It is often assumed that laboratory measured soil properties have both high accuracy and precision, and therefore any inaccuracy or imprecision is ignored. However, a recent interlaboratory comparison found considerable variability in lab measured soil property values among labs. For example, 6th FSCC interlaboratory comparison found that percent clay values had a coefficient of variation of 32% among 50 participating laboratories (Cools and Vos, 2010). Furthermore, our work found significant variation in soil texture measurements among 3 different soil laboratories in Ghana, with a CV of 47% for clay based on 10 soils with contrasting textures (2020, unpublished data). Such high imprecision complicates the ability to identify a true or standard value that can be used to evaluate accuracy. Based on these interlaboratory comparisons, it would be incorrect to simply assume a single laboratory-based texture class measurement represents the 'Gold Standard' for analysis of soil map accuracy. In fact, a recent study by Vos et al. (2016; 10.1016/j.geoderma.2015.12.022) that assessed the variance of field-based texture class estimates and their precision in determining a soil's particle size distribution, found that between 57-72% of deviations between field and laboratory derived texture estimates was due to laboratory measurement uncertainty and the fact that only texture classes were estimated in the field and not mass fractions.

So how do we deal with multiple sources of uncertainty? One approach is to determine the minimum level of measurement precision needed to inform a particular outcome. While soil particle size mass fractions (high measurement precision) are often required for soil modeling, soil texture classes (low measurement precision) are generally sufficient for on-farm soil management. Thus, using soil texture classes lowers our level of measurement precision which in turn minimizes the different sources of uncertainty (e.g., lab, field). Our use of the GAEZ soil suitability ratings and classes further lowered the measurement precision, which in turn further minimized the inherent uncertainty associated with lab and field-based measurements.

We have added additional text to the discussion in section "4.2.1 Sources of field sampling error" to expand on these points (lns 541-551).

> *'When evaluating soil data uncertainty, it is important to determine the minimum level of measurement precision need to inform a particular outcome. While soil particle size mass fractions (high measurement precision) are often required for soil modelling, soil texture classes (low measurement precision) are generally sufficient for on-farm soil management. Thus, using soil texture classes lowers our level of measurement precision which in turn minimizes the different sources of uncertainty (e.g., lab, field). Functional soil assessments often require even lower levels of measurement precision as demonstrated by the GAEZ framework where the soil suitability ratings and classes further lowered the measurement precision, which in turn further minimized the inherent uncertainty associated with lab and field-based measurements. Furthermore, in our evaluation of functional accuracy using the modified GAEZ framework, functional differences for soil texture only occur for textural classes either high in sand (sand, loamy sand, sandy loam) ($SQ_N$, Fig. 10) or high in clay (clay) ($SQ_W$). The higher accuracy of hand texture estimates in these regions of the soil texture space decreases the degree of sampling error in our functional accuracy evaluation.'*

**Reviewer Comment:**

*"The authors do attempt to account for the accuracy of the validation observation class labels using the results of several authors (Line 479) to argue that since sandy soils are often the most accurately identified and since most of the soils in the study are sandy then the validation observations are probably okay. The generalization of soil classes into soil suitability classes also helps address this issue; however, using the numbers provided by the authors may suggest that this is not an entirely robust assumption. For example, Salley et al., indicates that field technicians (and I assume that field technicians, not trained soil scientists collected the validation observations) are only able to correctly identify the true textural class 41% of the time (a rather dismal number), but that this improves to 78% if the adjacent textural class is accounted for. This suggests that the accuracy of the validation observations might be expected to be between 41% and 78%."*

**Author Response:**

We agree with Dr. Brungard that a 41% correct classification rate is rather dismal, but it is incorrect to assume that the 59% misclassification is solely attributable to field technician. We acknowledge that a large portion of this error is likely attributable to the field technician but we must also acknowledge that the laboratory measurements also have inherent inaccuracy. Widening the classification criteria to also include adjacent textural classes greatly improved the accuracy reported by Salley et al., (2019), and we also include this type of comparison in our analysis. Our study goes a step further by further generalizing our classification criteria by using the GAEZ soil suitability ratings and classes. Although this type of analysis was not evaluated in Salley et al., one would expect a further increase in accuracy well above 78%, and thus serves as a more reliable metric for accuracy evaluation given the inherent uncertainty associated with each data type (i.e., field vs. lab). See additional text addressing this issue on lines 541-551.

**Reviewer Comment:**

*"A more in-depth, if rather crude, analysis\* however suggests that the true accuracy of the class labels for the validation dataset is between 64% and 82%. Thus, if the true accuracy of the validation observations is 82% then even if the soil maps were 100% accurate (which they are not) then the maximum accuracy that the soil maps could achieve would*

*be 82%. This suggests that the reported accuracy metrics are not robust making a clear accuracy assessment difficult. Because the uncertainty of class labels in the validation data is not accounted for in the analysis I feel that the methods are not appropriate for this type of analysis."*

**Author Response:**
We agree with Dr. Brungard's analysis but wish to clarify two important points:
1. It is incorrect to assume that laboratory data is 100% accurate and that any difference between field and laboratory data is due to inaccuracy in field measurements. In the best-case scenario, field and laboratory data disagree in only 18% of cases, however based on previous work (including our own), some portion of this error is attributable to laboratory uncertainty. Previous studies have shown that certain soils, due to their minerology or chemical make-up, are not well-suited for laboratory particle-size analysis and field-based estimates may be considered more reliable than lab data (Landon, 1988). For example, with highly weathered oxide-rich tropical soils traditional laboratory techniques often underestimate clay content due to the soil's resistance to dispersion (Silva et al., 2015). These types of highly weathered and/or oxide rich soils (Plinthisols, Ferralsols, Acrisols) are common throughout our study area.
2. Second, the calculated accuracy range of 64% -82% for validation observations is based on an exact matching criteria. By widening the classification criteria to include adjacent classes or by using the GAEZ soil suitability ratings, the accuracy range for the validation observations will likely increase above the 64-82% accuracy which we feel is an acceptable level of accuracy given the inherent uncertainty associated with either type of validation data.

Additional text addressing this issue was added on lines 524-532.

**Reviewer Comment:**
*"There are several methods that might be employed to address the imprecision of the validation observations such as: 1) selecting a subset of the validation observations and running these through a lab (e.g., hydrometer) to verify class labels or 2) Use a method that accounts for measurement uncertainty (though I will admit to not knowing of a method to do such a comparison without retreating to a statistics textbook)."*

**Author Response:**
We appreciate these suggestions and agree that future work is needed to address issues relating to the uncertainty of validation data. But more work is needed to quantify the uncertainty of the validation data, both field and lab data. There needs to be greater recognition of the uncertainty associated with laboratory-based soil particle size analysis, which for certain soils can be quite high. Therefore, we feel that Dr. Brungard's first suggestion does not solve the issue of validation uncertainty. We agree with his second suggestion about using a statistical method that accounts for measurement uncertainty, and while some of the soil maps evaluated in this study contain measures of uncertainty that could be used in the validation procedure, this would also require a validation sample set with quantified uncertainty that would allow for the calculation of probability distribution functions for both the map and validation datasets. Our solution to the uncertainty problem was to base our evaluation on several sets of validation labels with increasing degrees of generality (i.e., soil texture class, soil texture groups [includes adjacent classes], GAEZ suitability). While the 'true' accuracy of each set of validation labels is unknow, we can infer (as Dr. Brungard did for the soil texture class labels) an estimate of accuracy based on previous studies. As we stated earlier, it is important to understand the level of precision required to address a particular outcome, and in the case of agricultural management of

rainfed maize, the GAEZ suitability labels provide an appropriate level of precision and inferred accuracy. While the accuracy of validation labels based on soil texture classes may be too low (64% -82%) to reliably evaluate the map sources, we feel that the GAEZ labels are sufficiently accurate and corroborate the results from the other labels.

Based on Dr. Brungar's comments we have added additional text to manuscript to clarify our stance on validation uncertainty and the use of field-based validation data:

> *"Furthermore, some soils due to their minerology or chemical make-up are not well-suited for laboratory particle-size analysis and field-based estimates may be considered more reliable than lab data (Landon, 1988). This is true for highly weathered oxide-rich tropical soils where traditional laboratory techniques often underestimate clay content due to the soil's resistance to dispersion (Silva et al., 2015). However, difficulties with lab-based clay estimation extend beyond oxide-rich tropical soils, as was shown in the 6[th] FSCC interlaboratory comparison which found that clay content was one of the most difficult properties to consistently measure, with a coefficient of variation (CV) of 32% among 50 participating laboratories (Cools and Vos, 2010). A recent interlaboratory comparison among three soil laboratories in Ghana revealed significant variation in soil texture measurements, with a CV of 47% for clay based on 10 soils with contrasting textures (2020, unpublished data)."*

> *Cools, N. and Vos, B. De: 6 th FSCC Interlaboratory Comparison 2009 Further development and implementation of an EU-Level Forest Monitoring System ( FutMon ), Life + Regulation of the European Commission , in cooperation with the International Cooperative Programme on Assessment a, , 32(0), 2010.*

> *Landon, J. R.: Towards a standard field assessment of soil texture for mineral soils, Soil Surv. L. Eval., 8(3), 161–165, 1988.*

> *Silva, J. H. S., Deenik, J. L., Yost, R. S., Bruland, G. L. and Crow, S. E.: Improving clay content measurement in oxidic and volcanic ash soils of Hawaii by increasing dispersant concentration and ultrasonic energy levels, Geoderma, 237, 211–223, doi:10.1016/j.geoderma.2014.09.008, 2015.*

**Reviewer Comment:**
*"I am also uncomfortable with the conclusions. The authors state "results from this study highlight the need for on-site verification technologies… that can constrain the… site-based soil map predictions". I generally agree with this, but feel that this conclusion focuses too much on such technologies. What about improving the soil maps? If soil maps were 100% accurate then they would be a very useful source of soil information and we wouldn't need on-site verification technologies. I realize that no soil map (or any map) will every be 100% accurate, but the authors should consider this in their conclusions. Also, what about training more soil scientists? A broader cadre of local professional soil scientists could provide such site-specific information and might be more familiar with local soils and issues."*

**Author Response:**
We thank Dr. Brungard for this insightful comment and we fully agree with these additional suggestions for improving the utility of soil maps for farm scale management. We have revised our conclusions to include these valuable suggestions (lns. 690- 706).

*"Results from this study highlight the variable site-specific accuracy of current soil map information and the potential implications for on-farm decision making. The urgent need for reliable soil information, that is, information with a specified accuracy and precision for a targeted objective (e.g., attainable crop yield), has become increasing clear and many areas of research are being advanced to address this global challenge. Among these is the continued improvement of soil maps, particularly through the advancement of predictive soil mapping technologies, including improved predictive algorithms, expanded soil training/testing datasets, and advancements in the quantification of model uncertainty. For example, both predictive soil maps evaluated in this study (SoilGrids, iSDA) provide uncertainty maps and future work is needed to utilize this information for large-scale site-specific analysis. There is also a need for the training of more soil scientists to expand the characterization and sampling of soil landscapes with high information uncertainty resulting from a lack of ground-truthed samples and/or poorly understood soil landscape relationships. Profession soil scientists can also assist in the training of non-soil specialists (e.g., field enumerators, citizen scientists), who with the help of on-site verification technologies like smartphone-based decision support tools (e.g., LandPKS) and proximal sensors (e.g., VisNIR) can collect accurate field-based soil data. This information can then be used both directly to inform site-specific decision making (e.g., smallholder fertilizer application rates), as well as to improve soil map predictions, as demonstrated by the use of LandPKS field data in iSDA soil map predictions. All these advancements can help constrain the uncertainty associated with site-based soil map predictions and help provide access to accurate soil property information so urgently needed by smallholder farmers to improve soil health and enhance the long-term production capacity of their soils."*

**Technical Corrections:**

1. *Figure 5 and 5. A-d. I believe this would be much more helpful if these were to show the resampled depths instead of the original depths. Also, do i-n show M2F observations at the pedon scale or farm scale?*
   **Response:** We have updated this figure to show resampled depths for the original maps. Panels i-n show M2F observations at point-support. This has been added to the figure caption.

2. *iSDAsoil soil maps are generated from a suite of ensemble models. My own application of ensemble modeling techniques suggests that they can miss the extreme values (although this observation was likely a result of the specific ensemble method that I tested). It might be more informative to use the predictions from Hengl et al 2015. https://journals.plos.org/plosone/article?id=10.1371/journal.pone.0125814 If the authors want to test ALL available maps for Ghana, then this would be good to include, but I leave this to the authors discression.*
   **Response:** This is an interesting finding regarding ensemble modeling and while we agree it would be interesting to test for any differences between SoilGrids v1 and v2, our goal was to test current soil map data sources likely to be used for soil decision making. Since SoilGrids v2 supersedes v1, we only included the most recent data. We added additional text to the discussion addressing this issue in section 4.2.2 on lines 756-774.

   > *"In predictive soil modelling, model error is composed of two components: bias, which relates to model accuracy; and variance, which relates to model precision or uncertainty. SoilGrids and iSDA maps both employ ensemble modelling approaches to calculate spatial predictions of model uncertainty (SoilGrids: Quantile Random Forest; iSDA: Ensemble*

*Bootstrapping). Ensemble models are effective at increasing model accuracy and are often implemented using some form of model averaging (Polikar, 2012). Assuming the different soil models produce different errors at each location, averaging the model outputs generally reduces the error (model bias) by averaging out the error components. A downside of model averaging is that it has a smoothing (variance-reducing) effect which can remove valid information from the outer ranges of the soil property distribution. There are many cases where the ability to predict these 'extreme' values is crucial, for example, at the smallholder farm scale where the risk or cost associated with incorrectly identifying soil constraints can be high for cash-constrained farmers. Depending on the soil management scenario, there are financial costs associated with both false negative results (i.e., failure to detect constraint – Type 2 error; e.g., failure to lime a very strongly acidic soil before applying fertilizer) and false positive results (i.e., false detection of constraint – Type 1 error; e.g., applying lime to a neutral soil). While the mean or median predicted soil values may not indicate the presence of soil constraints, spatial predictions of model uncertainty can be used to determine where constraints have a predicted probability of occurrence. Future research is needed to evaluate information on soil map uncertainty and how this information can be effectively communicated and incorporated into smallholder agronomic decision making."*

3. *Please explain the balanced error rate in more detail. Is this (UA + PA)/2 or some other equation? What does this mean? Please also clarify that the error rate seems to be an error metric with the users and producer's accuracy rates are accuracy metrics.*
   **Response:** The balanced error rate (BER) is the average of the errors in each property class. This includes both the errors of omission or false negative rate (FNR) (i.e., [*False positive/(False positive + True negative)*]]) and the error of commission or false positive rate (FPR) (i.e., [*False positive/(False positive + True negative)*]]). BER is calculated as: ½(FPR +FNR). BER is useful because it accounts for both types error, Type I and II errors. When datasets have high class imbalance the overall error rate is not very informative. Using the average of the FNR and FPR, BER can account for problems of class imbalance, where models that overpredict the dominant class will receive a higher BER. We added additional text to the methods on lines 243-247:

   *"The balanced error rate (BER) is the average of the errors in each property class which includes both the error of omission or false negative rate (FNR) (i.e., [False positive/(False positive + True negative)]) and the error of commission or false positive rate (FPR) (i.e., [False positive/(False positive + True negative)]). BER is calculated as: (FPR +FNR)/2. Using the average of the FNR and FPR, BER can account for problems of class imbalance, where models that overpredict the dominant class will receive a higher BER."*

4. *The paragraphs from lines 255-265 seem duplicated. Please revise.*
   **Response:** We have revised this section.

5. *Please put the discussion of SQ ratings and associated equations in their own paragraph as it is confusing as currently written.*
   **Response:** We have separated this section into its own paragraph.

6. *Line 339: How do you know the delineation procedure captured 48% of the fields area if you do not have farm field maps? (the lack of such maps was stated earlier as the justification for the delineation proceedure).*
   **Response:** It is correct that we do not have delineated farm boundaries but we do have two pieces of information: (1) the total farm area, and (2) the locations of three points within the farm boundary. By creating a convex hull (in our case a triangle since we only have three points) and applying a narrow buffer (10m), we were able to delineate an area within each field. Based on the known area of each field we were then able to compare our delineated area to the total field area, which on average captured 48% of the fields area. This is described in the methods section (2.5 Soil map accuracy assessment at field-support).

7. *Line 509: ... of the point soil data...*
   **Response:** Correction was made.

8. *Line 525: scale does not really translate well to discussing gridded variables. I think that the neighborhood size, or distance-over, is a more accurate description. Also, maybe a salient point to this discussion: the source of the geospatial source data (DEM's in particular) is important. A 30m DEM from contours will be much different than a 30m DEM derived from upscaled LiDAR.*
   **Response:** For gridded variables scale refers to both the spatial extent or neighborhood size and the grid resolution. We have clarified this point. We agree that the accuracy and precision of the source data is important but our main point here is that increasing spatial resolution (i.e., grid size) doesn't automatically produce a more accurate map due to scaling effects.

9. *Please check the following sentences for grammatical errors: 48, 89, 91, 308, 331, 369, 370, 591*
   **Response:** Grammatical errors have been corrected.

10. *Line 153 should be grouped with the preceding paragraph.*
    **Response:** We have incorporated this change (lns 150-151).

11. *Please check references. Lines 169 and 172 are missing, also Ritchie and Roser does not seem complete.*
    **Response:** References have been added and updated

12. *Line 230, what about user's accuracy?*
    **Response:** We have corrected this omission.

**Response to anonymous reviewer (RC2)**

**Reviewer Comment:**
*"This manuscript compares soil maps (conventional and digital soil mapping) in terms of their prediction accuracies and their accuracies for soil suitability assessment for maize farms across Ghana. It aims to assess whether these soil maps could provide useful information for guiding farm management. This is an important question, and the manuscript is well written and provides a useful analysis, finding limitations in the ability of the soil maps to predict constrained soils. I have a few suggestions and queries below, but overall would think it could be published with some revisions."*

*"Section 2.5 describes field-support validation of predictions and I like the fact that this is being done, but am not convinced by the details of how it has been applied. The described process seems to be (for each field) first inferring a field boundary, second extracting from the soil map the predictions from all pixels within the field boundary, and third averaging these predictions; the result is compared with the average of the three sampled soil data from the field. But given your field-support validation data here are the average of three known points, shouldn't your associated prediction be the average of the predictions (as extracted from the soil maps) at these three same points? This could still be referred to as 'field-support validation', but would be comparing like with like, rather than two slightly different definitions of 'field support'. (While I don't expect this to make a huge difference to your results, I think it would be a more justifiable approach.)*

**Author Response:**
The reviewer poses an important question about how to conduct a field-support validation. The intent of validation at a field-support is to estimate and evaluate the average soil property values within a field. Thus, validation samples are collected that represent the field average, and the map accuracy is evaluated within the entire area contained within the field. Since soil maps have different spatial support (ISDA=30m, SoilGrids=250m), the number of predicted values intersected by a field will vary, but the goal is to obtain an average predicted value for each field. If we were to only take the predicted value at each validation point, the average estimated field value would not reflect the average value predicted by the map. The figure below (Fig.1) illustrates this, which shows a map of predicted clay percentage (0-10 cm) from the ISDA map in southern Ghana. The three small black circles are validation point that have a clay surface texture (0-10 cm). If we were to only take the pixels that intersected our validation points, we would conclude that the map and the validation points agreed at the field-scale. Whereas, when we take the average value for the field (i.e., all pixels within field, all points within field), we see that the map predictions are not in agreement with the validation data. For additional clarification on conducting validation at different spatial support, we direct the reviewer to the following papers:
https://bsssjournals.onlinelibrary.wiley.com/doi/full/10.1111/sum.12694,
http://dx.doi.org/10.1016/j.geoderma.2014.11.026.

[Figure]

*Figure 1. ISDA map of predicted clay percentage (0-10 cm). Small black circles show location of validation sites with clay surface textures (0-10). For reference, a clay textural class requires >=40% clay.*

**Reviewer Comment:**

*"Line 237: Is the Balanced error rate the (unweighted) average of the Producer's accuracies for the classes? I'm not convinced/sure about what this measure is saying – if there is a class (e.g. silty clay/sandy clay) with very few data (presumably because it is quite rare in the region of interest), won't the balanced error rate place too much importance on predictions of this class? Perhaps add a sentence after line 237 to clarify why use the balanced error rate. (I could see why a BER as a weighted average of PA values could be useful, for instance where you know that for some reason some classes were over-represented in your data, and had some knowledge of what the real proportions of the classes across the region of interest should be – but to use equal weights in the averaging seems to me to be saying that your initial expected proportions were equal for all classes, which seems unlikely to me.)"*

**Author Response:**

The balanced error rate (BER) is the average of the errors in each property class. This includes both the errors of omission or false negative rate (FNR) (i.e., [*False positive/(False positive + True negative)*])) and the error of commission or false positive rate (FPR) (i.e., [*False positive/(False positive + True negative)*]). BER is calculated as: ½(FPR +FNR). BER is useful because it accounts for both types error, Type I and II errors. When datasets have high class imbalance the overall error rate is not very informative. Using the average of the FNR and FPR, BER can account for problems of class imbalance, where models that overpredict the dominant class will receive a higher BER. We added additional text to the methods on lines 243-247:

*"The balanced error rate (BER) is the average of the errors in each property class which includes both the error of omission or false negative rate (FNR) (i.e., [False positive/(False positive + True negative)]) and the error of commission or false positive rate (FPR) (i.e., [False positive/(False positive + True negative)]). BER is calculated as: (FPR +FNR)/2. Using the average of the FNR and FPR, BER can account for problems of class imbalance, where models that overpredict the dominant class will receive a higher BER."*

**Reviewer Comment:**

*"I wonder if another test might be helpful to provide further context about the quality of predictions from the maps. With the dataset that has 3 points sampled in each field, you could test how accurate predictions would be if you used just one of the sampled points from the field as 'representative' of the field, and evaluated how accurate this was at predicting the two other points in the field (to evaluate how good management would be based on a single 'representative' sample from the field). If there is a lot of within-field variation, even this might give poor prediction accuracies. This might add further insight into what is written on line 562 – "…fail to meet the accuracy requirements…" – would the use of a sample from the field also fail?"*

**Author Response:**

The reviewer poses an interesting idea and we agree that in-field variability is an important consideration for field sampling but this was beyond the scope of this project. This would be an interesting analysis for future work.

**Reviewer Comment:**

*"Line 431: I can't see the numbers you are referring to here in Table 2, they seem to me to be a lot higher than this – from my interpretation of the numbers in Table 2, it seems that OA-adj was over 80% in many cases (eg iSDA, OA-adj, soil texture: 0.9). Also check the numbers on line 593, and check the numbers throughout."*

**Author Response:**

Yes, these numbers were incorrectly reported. We have reviewed and corrected all numbers reported in the text.

**Reviewer Comment:**

*"Regarding everything being predicted as 'no constraint' (eg by iSDA, Fig 8) – I think that predicting the more extreme values of soil constraints is always going to be difficult, and the 'expected value' as extracted from the soil map will rarely give the extreme values – I guess a bit like predicting rare events. A possible point for discussion is that you could use the map of predictions + its uncertainty to give (for each pixel/prediction location) a probability of constraint, and this could be a more appropriate way of informing management decisions. (Although this would very rarely be done in practice, could note that tools to implement this type of analysis of the soil map + its uncertainty map could be something worth looking at in future?)"*

**Author Response:**

This is a very good point and the idea to use the prediction uncertainty is a good one. We agree that predicting the extremes can be challenging, especially when dealing with global soil maps. However, the risk or cost associated with incorrectly identifying these soil constraints can be extremely high at the smallholder scale. There are financial costs associated with both false negative results (i.e., failure to detect constraint – Type 2 error) and false positive results (i.e., false detection of constraint – Type 1 error). The uncertainty information could help with this type of risk analysis. We agree that this type of analysis is worth looking at in the future and we have added this point to the discussion on lines 579-597:

> *"In predictive soil modelling, model error is composed of two components: bias, which relates to model accuracy; and variance, which relates to model precision or uncertainty. SoilGrids and iSDA maps both employ ensemble modelling approaches to calculate spatial predictions of model uncertainty (SoilGrids: Quantile Random Forest; iSDA: Ensemble Bootstrapping). Ensemble models are effective at increasing model accuracy and are often implemented using some form of model averaging (Polikar, 2012). Assuming the different soil models produce different errors at each location, averaging the model outputs generally reduces the error (model bias) by averaging out the error components. A downside of model averaging is that it has a smoothing (variance-reducing) effect which can remove valid information from the outer ranges of the soil property distribution. There are many cases where the ability to predict these 'extreme' values is crucial, for example, at the smallholder farm scale where the risk or cost associated with incorrectly identifying soil constraints can be high for cash-constrained farmers. Depending on the soil management scenario, there are financial costs associated with both false negative results (i.e., failure to detect constraint – Type 2 error; e.g., failure to lime a very strongly acidic soil before applying fertilizer) and false positive results (i.e., false detection of constraint – Type 1 error; e.g., applying lime to a neutral soil). While the mean or median predicted soil values may not indicate the presence of soil constraints, spatial predictions of model uncertainty can be used to determine where constraints have a predicted probability of occurrence. Future research is needed to evaluate information on soil map uncertainty and how this information can be effectively communicated and incorporated into smallholder agronomic decision making. "*

---

## Author Response (AR3)

Dear Editor,

Thank you for the opportunity to revise our manuscript. We have addressed the two minor revisions requested by Reviewer 2.

Kind regards,

Jonathan

**Response to anonymous reviewer (RC2)**

1) *RC2: The sentence on lines 344-345 doesn't make sense, please check.*

   AR: We revised this sentence: *"We limited the calculation of SQs to the soil properties common among all five soil sources, which were soil texture, rock fragments, and soil depth."*

2) *RC2: Line 461 : I was quite confused by this sentence on first reading. I think these accuracies represent estimated agreement between field texture class data and (unknown) actual field texture classes of the evaluation datasets, using the weighted mean as per lines 270 – 275. I think a slight rewording from 'Estimated accuracies' to perhaps 'Estimated agreement between … and …' could help the reader here, and perhaps a reminder/pointer to the part of the method section to refer to.*

   AR: Thank you for this suggestion. We have adopted the suggested terminology to help clarify this section. *"Due to potential inaccuracies in field estimated soil texture classes, we estimated the agreement between field and laboratory measured texture classes using our global meta-analysis of soil field texture measurement accuracy. Estimated agreement (± standard deviation) between field and laboratory values was 58 ±10% and 53 ±11% for the FTF-M2F and M2F evaluation datasets, respectively. Estimated agreement increased to 90 ±5% for FTF-M2F and 86 ±11% for M2F when allowing for class adjacency (OAadj)."*